# The Concept of Indeterminable NASH Inducted by Preoperative Diet and Metabolic Surgery: Analyses of Histopathological and Clinical Features

**DOI:** 10.3390/biomedicines10020453

**Published:** 2022-02-15

**Authors:** Akira Sasaki, Akira Umemura, Kazuyuki Ishida, Naoto Takahashi, Haruka Nikai, Hiroyuki Nitta, Yasuhiro Takikawa, Keisuke Kakisaka, Tamami Abe, Masao Nishiya, Tamotsu Sugai

**Affiliations:** 1Department of Surgery, Iwate Medical University, Iwate 028-3695, Japan; aumemura@iwate-med.ac.jp (A.U.); takanao@iwate-med.ac.jp (N.T.); hnikai@iwate-med.ac.jp (H.N.); hnitta@iwate-med.ac.jp (H.N.); 2Department of Diagnostic Pathology, Dokkyo Medical University, Tochigi 321-0293, Japan; ishida-k@dokkyomed.ac.jp; 3Division of Hepatology, Department of Internal Medicine, Iwate Medical University, Iwate 028-3695, Japan; ytakikaw@iwate-med.ac.jp (Y.T.); keikaki@iwate-med.ac.jp (K.K.); atamami@iwate-med.ac.jp (T.A.); 4Department of Pathology, Iwate Medical University, Iwate 028-3695, Japan; nishiya@iwate-med.ac.jp (M.N.); tsugai@iwate-med.ac.jp (T.S.)

**Keywords:** NAFLD, NASH, liver fibrosis, hepatocyte ballooning, metabolic surgery

## Abstract

Practitioners routinely perform intraoperative liver biopsies during laparoscopic sleeve gastrectomy (LSG) to evaluate nonalcoholic fatty liver disease (NAFLD). In some patients, hepatocyte ballooning, inflammation, and fibrosis without steatosis are observed, even in the absence of other etiologies. We call this finding indeterminable nonalcoholic steatohepatitis (Ind-NASH). In this study, we clarified the prevalence, as well as histopathological and clinical features, of Ind-NASH through intraoperative liver biopsy in Japanese patients presenting with severe obesity. We enrolled 63 patients who had undergone LSG and intraoperative liver biopsy. In patients diagnosed with histopathological NASH, we performed protocol liver biopsies at 6 and 12 months after LSG. We statistically analyzed these histopathological findings and clinical parameters and found the prevalence rate of Ind-NASH discovered through intraoperative biopsy to be 15.9%. Protocol liver biopsy also revealed that Ind-NASH was an intermediate condition between NASH and normal liver. The clinical features of patients with Ind-NASH are a higher body weight compared to NASH (134.9 kg vs. 114.7 kg; *p* = 0.0245), stronger insulin resistance compared to nonalcoholic fatty liver (homeostasis model assessment–insulin resistance: 7.1 vs. 4.9; *p* = 0.0188), and mild liver dysfunction compared to NASH. Patients with Ind-NASH observed positive weight-loss effects from a preoperative diet compared to the postoperative course (percentage total weight loss: 32.0% vs. 26.7%; *p* < 0.0001). Patients with Ind-NASH may also be good candidates for metabolic surgery owing to their good treatment response; therefore, efforts should be made by specialists in the near future to deeply discuss and define Ind-NASH.

## 1. Introduction

The incidence of nonalcoholic fatty liver disease (NAFLD) is rapidly increasing, and it is one of the most prevalent chronic liver diseases stemming from hepatic virus infection worldwide [1]. NAFLD has been recognized as resulting from an unhealthy lifestyle and a lack of daily exercise; therefore, NAFLD is associated with metabolic dysfunction such as obesity, type 2 diabetes (T2D), and metabolic syndrome [2]. NAFLD encompasses a spectrum of diagnoses, including simple steatosis, nonalcoholic steatohepatitis (NASH), liver cirrhosis, and hepatocellular carcinoma [3]. Obesity and increased body mass index (BMI) correlate with the risk of NAFLD development. In Japanese patients with severe obesity, the prevalence of NASH is reportedly 77.5% [4]. We also reported that the prevalence rate of liver fibrosis in patients who had undergone laparoscopic sleeve gastrectomy (LSG) as a metabolic surgery (MS) procedure was 63.2% [5].

For patients with severe obesity, metabolic surgeries are the most effective weight-loss therapies, and bariatric procedures can reduce the severity of various diseases (e.g., T2D, hypertension, dyslipidemia, and obstructive sleep apnea) [6,7,8]. With regard to the use of MS for NAFLD/NASH, many reports have highlighted the resolution of NAFLD/NASH following MS. We also observed significant histopathological improvements through sequential liver biopsies (including those pertaining to steatosis, lobular inflammation, hepatocyte ballooning, and NAFLD activity score) from the intraoperative period to two years after LSG [5,9].

Histopathological examination is the most reliable modality by which to diagnose NASH, as steatosis, hepatocyte damage, inflammation, and fibrosis are clearly evaluated in detail. Currently, the fatty liver inhibition of the progression (FLIP) algorithm is the key point by which to distinguish whether a case is NASH [10]. In the FLIP algorithm, if steatosis is <5%, neither inflammation nor liver fibrosis are considered for NASH. When performing LSG as MS, we routinely performed an intraoperative liver biopsy to evaluate the histopathological findings. However, intraoperative liver biopsy revealed some cases of hepatocyte ballooning, inflammation, and fibrosis, or steatosis, without other etiologies (Figure 1). These histopathological findings have not yet been clearly defined, owing to a lack of histopathological findings, clinical features, and exhaustive discussion. In the present study, some patients did not have sufficient steatosis but did have clear inflammation and/or fibrosis that resembled that in NASH before preoperative weight loss. We term this histopathological finding “indeterminable NASH” (Ind-NASH).

In the present study, we clarified the prevalence and histopathological and clinical features of Ind-NASH via intraoperative liver biopsy in Japanese patients with severe obesity. We also investigated changes in histopathological findings in patients with histopathological NASH, as discovered through postoperative ultrasound-guided liver biopsies at 6 and 12 months after LSG. Furthermore, we define the term “indeterminable NASH” as a concept of the histopathological condition of returning from NASH to normal liver.

## 2. Materials and Methods

### 2.1. Patients

LSG procedures have been covered by Japan’s health insurance system since 2014. In 2008, we started to perform LSG for Japanese patients with severe obesity, with 98 cases of LSG performed as of September 2020. All the patients met the following inclusion criteria for LSG treatment, as established by Japanese insurance practices: 18–65 years of age, severe obesity (BMI ≥ 35 kg/m^2^), and the presence of at least one comorbidity with resistance to medical treatment (e.g., hypertension, T2DM, dyslipidemia, or obstructive sleep apnea). The exclusion criteria were a history of alcohol abuse, secondary obesity (drug-induced or due to endocrine disease), and the presence of major psychiatric disorders [11]. In this study, we enrolled 63 qualified patients (Figure 2). In patients with histopathological NASH, intraoperative protocol liver biopsies were performed at 6 and 12 months after LSG. When the histopathological remission of NASH was achieved, further protocol liver biopsy was not performed for the same patient.

This study was approved by the institutional ethics committee of Iwate Medical University Hospital (approval number: H27-47, accessed on 6 August 2015) and conducted according to the ethical principles of the Declaration of Helsinki. We obtained informed consent from each patient before enrollment, and patient anonymity was strictly protected.

### 2.2. Treatment

Patients started a very low calorie preoperative diet following their initial visit to our clinic. Although the degree of preoperative weight loss varied depending on the severity of obesity-related disease, the waiting period, initial body weight, and the timing of LSG were set to when the preoperative total weight-loss percentage (%TWL) was 5% or higher. With regard to surgical procedures, each LSG procedure involved a 70–80% gastric-volume reduction undertaken by resecting the stomach alongside a 36-French esophagogastroduodenoscopy, beginning 4 cm from the pylorus and ending at the angle of His. All patients were continuously evaluated and cared for by a multidisciplinary team, from initial visit to postoperative follow-up, in order to improve weight-loss and metabolic effects.

### 2.3. Data Collection

#### 2.3.1. Weight-Loss Effects

For each enrolled patient, we evaluated the clinical data and weight-loss effects at baseline and 12 months after LSG. As weight-loss parameters, body weight, BMI, %TWL, waist, visceral fat area (VFA), and subcutaneous fat area (SFA) measurements were recorded and calculated. VFA and SFA were measured using 64-row computed tomography (CT) (Aquilion^TM^; Toshiba Medical Systems Corporation, Tokyo, Japan) at the umbilicus-level single slice. The SFA and VFA parameters were measured at the same time and are expressed in cm^2^.

#### 2.3.2. Metabolic Parameters

Using a prospectively registered database, we examined the following data collected at the first visit, 6 months after LSG, and 12 months after LSG, with regard to diabetic parameters: fasting blood glucose (FBG), immunoreactive insulin (IRI), hemoglobin A1c (HbA1c), C-peptide, homeostatic model assessment of insulin resistance (HOMA-IR), and homeostasis model assessment for beta cell function (HOMA-β). We also collected data on NASH parameters: total cholesterol (TC), triglyceride (TG), low-density lipoprotein cholesterol (LDL-C), high-density lipoprotein cholesterol (HDL-C), aspartate aminotransferase (AST), alanine transaminase (ALT), γ-glutamyltranspeptidase (GGT), albumin, total bilirubin (T-Bil), ferritin, type-4 collagen 7S, hyaluronic acid, malondialdehyde-modified low-density lipoprotein (MDA-LDL), transferrin, and plasminogen activator inhibitor-1 (PAI-1).

Using laboratory data, we calculated FIB-4 index values [12], NAFIC scores [13], and NAFLD fibrosis scores [14]. In addition, we also measured liver volume using CT volumetry. To calculate the liver-to-spleen (L/S) ratio, we measured hepatic and splenic CT attenuation values on non-contrast CT using 20 circular region-of-interest cursors in the liver and spleen. We obtained all measurements in the region of uniform parenchymal CT attenuation. To measure VFA, SFA, liver volume, and L/S ratio, we downloaded CT images as digital imaging and medical files into a computer workstation (SYNAPSE VINCENT v6.1; Fujifilm, Tokyo, Japan). We concurrently undertook CT examinations.

To exclude other liver diseases, such as viral hepatitis, autoimmune hepatitis, and alcoholic hepatitis, we performed screening laboratory examinations (HBs-antigen, HBs-antibody, HBc-antibody, HCV-antibody, antinuclear antibody, smooth muscle antibody, and immunoglobulin G4) and detailed medical consultation at the first visit.

#### 2.3.3. Histopathological Evaluation of Liver Biopsy

Due to the risks associated with preoperative liver biopsy, all enrolled patients underwent intraoperative liver biopsies. Specimens were formalin-fixed and stained with hematoxylin–eosin, silver reticulin, and Masson trichrome to evaluate liver fibrosis. All specimens were evaluated by multiple experienced pathologists blinded to the clinical data of the patients. The histopathological diagnosis of NASH was made using the use of the FLIP algorithm [10], the NAFLD activity score [15], and the Matteoni classification [16]. Grading and staging were evaluated according to the Brunt classification [17]. Steatosis was evaluated in terms of the percentage of hepatocytes affected by steatosis. The NAFLD activity score was determined as the sum of the following scores: steatosis (0–3), lobular inflammation (0–3), and ballooning (0–2) (thus, the range of the total possible score for any individual was 0–8).

We diagnosed Ind-NASH in any patient with histopathological findings of any inflammation, hepatocyte ballooning, or liver fibrosis with steatosis <5%. If there was liver fibrosis in Ind-NASH, previous classifications were not applied to Ind-NASH; for such cases, we used another scoring system for liver fibrosis, called the pericellular fibrosis score (PFS) [5]. This score reflects the extent of pericellular fibrosis around the central veins, as follows: no fibrosis (PFS 0), pericellular fibrosis confined to the proximity of central veins and present in <50% of central veins (PFS 1), pericellular fibrosis confined to the proximity of central veins and present in 50% or more of central veins (PFS 2), and pericellular fibrosis around the central veins with periportal fibrosis or bridging fibrosis (PFS 3) (Figure 3) [5].

### 2.4. Statistical Analysis

The data are presented numerically, with percentages for categorical variables and means ± standard deviations for continuous variables. We performed statistical analysis using chi-squared tests for categorical variables and a Steel test, Student’s *t*-test, or Mann–Whitney *U* test for continuous variables. We processed univariate and multivariate analyses using a logistic regression model. First, we performed univariate logistic regression analysis with several preoperative and postoperative factors affecting histopathological liver improvement. We then simultaneously incorporated significant variables (*p* < 0.15 or *p* < 0.01) into the logistic regression model, along with some significant factors. To calculate the cut-off value, we used a receiver operating characteristic (ROC) curve and calculated the area under the curve (AUC) to evaluate accuracy. In all the statistical analyses, any *p*-value lower than 0.05 was considered statistically significant. All the statistical analyses were performed using JMP Pro v14 (SAS Institute Inc., Cary, NC, USA).

## 3. Results

### 3.1. Baseline Characteristics of the Enrolled Patients

Table 1 lists the baseline characteristics of all the patients in this study. There were no patients with other chronic liver diseases, e.g., alcohol abuse, during the initial screening examination. The average initial body weight and BMI were 119.1 kg and 43.6 kg/m^2^, respectively. Forty of the enrolled patients had T2D, and we stratified these patients in terms of histopathological diagnoses of intraoperative liver biopsy. Intraoperative liver biopsy revealed that 35, 18, and 10 patients were diagnosed with NASH, nonalcoholic fatty liver (NAFL), and Ind-NASH, respectively; we hence classified these patients into three groups (NASH, NAFL, and Ind-NASH). In this intergroup comparison, body weight in the Ind-NASH group was significantly higher than that in the NASH group (134.9 kg vs. 114.7 kg; *p* = 0.0245). In contrast, insulin resistance was significantly worse in the NASH group than in the NAFL group (HOMA-IR; 7.1 vs. 4.9). Liver dysfunction was significantly worse in the NASH group than in either of the other groups; the L/S ratio was significantly higher in the Ind-NASH group, owing to it having less steatosis relative to the NASH group (1.0 vs. 0.8; *p* = 0.037). Figure 4 presents the clinical features of Ind-NASH, along with significant parameters.

In addition, preoperative diet periods and weight-loss effects were compared. The preoperative diet period of NASH, NAFL, and Ind-NASH groups were 80.1 ± 82.5 days, 158.4 ± 251.0 days, and 61.8 ± 41.2 days, respectively, with no significant differences. The preoperative weight-loss (21.6 ± 12.9 kg vs. 9.6 ± 5.7 kg, *p* = 0.0167) and %TWL (15.3 ± 6.2% vs. 8.3 ± 4.4%, *p* = 0.0058) in the Ind-NASH group were significantly better than those of the NASH group. On the other hand, the preoperative reductions in the liver volumes of NASH, NAFL, and Ind-NASH groups were 384.2 ± 388.6 mL, 493.8 ± 557.9 mL, and 383.8 ± 389.2 mL, respectively, with no significant differences. According to these findings, liver volume equally decreased in each group; however, the residue of liver steatosis remained uneven.

### 3.2. Weight-Loss and Metabolic Effects after LSG in Each Group

Almost all parameters dramatically improved 12 months after LSG, except for type-4 collagen 7S (*p* = 0.4225), hyaluronic acid (*p* = 0.2973), and transferrin (*p* = 0.1621) (Table 2). The %TWL of all patients 12 months after LSG was 26.7%. In the NASH group, the %TWL was 25.7%, and HOMA-β (*p* = 0.1342), γ-GTP (*p* = 0.1456), and LDL-C (*p* = 0.1038) did not significantly improve. In the NAFL group, the %TWL was 25.9%, and HOMA-β (*p* = 0.0751), LDL-C (*p* = 0.2349), and MDA-LDL (*p* = 0.1277) did not significantly improve (Table 2).

In the Ind-NASH group, the %TWL was 32.0%. On the other hand, unlike in the NASH and NAFL groups, the metabolic parameters of T2D and NASH did not improve in the Ind-NASH group.

### 3.3. Histopathological Changes along with Protocol Liver Biopsy

#### 3.3.1. Patient Flowchart along with Protocol Liver Biopsy

Intraoperative liver biopsy revealed that 35 patients (55.6%) had histopathological NASH. In total, 10 patients (15.9%) had Ind-NASH, and no patients had normal livers. Protocol liver biopsy at 6 months after LSG revealed that four patients were able to achieve histopathological NASH remission. On the other hand, patients with Ind-NASH showed the second most dominant histopathological finding in contrast to intraoperative liver biopsy. At 12 months after LSG, 11 patients achieved histopathological remission. Furthermore, a normal liver became the first dominant histopathological finding. Histopathological NASH returned to normal livers in 15 patients (42.8%) within 12 months following LSG (Figure 5).

#### 3.3.2. Histopathological Findings of the Protocol Liver Biopsy

Figure 6 presents a typical successful LSG case. This case was of a 47-year-old man with an initial body weight and BMI of 152.8 kg and 53.6 kg/m^2^, respectively. Intraoperative liver biopsy of the case revealed the following histopathological findings: steatosis was approximately 20% with Matteoni type 4; total NAFLD activity score was 2 (steatosis: 1; lobular inflammation: 1; ballooning: 0); PFS was 1; and Brunt stage and grade were 1 and 1, respectively. From these findings, we were able to render a diagnosis of NASH. Protocol liver biopsy at six months after LSG revealed the following histopathological findings: steatosis was <5%, total NAFLD activity score was 1 (steatosis: 0; lobular inflammation: 1; ballooning: 0), and PFS was 1; hence, the diagnosis was Ind-NASH. Histopathological remission occurred 12 months after LSG.

With regard to steatosis changes, the percentages of steatosis significantly improved at 6 months (7.5% vs. 25.2%; *p* < 0.05) and 12 months after LSG (8.0% vs. 25.2%; *p* < 0.05), compared to intraoperative liver biopsy (Figure 7).

However, other histopathological parameters remained unchanged. There was no significant change in PFS, or in Brunt grading and staging. However, the Matteoni type and NAFLD activity scores (all components and total) significantly improved 12 months after LSG (Figure 8). These results suggest that it might take up to 12 months before an improvement in liver fibrosis is observed in NASH patients.

### 3.4. Prognostic Factors for the Improvement of NASH and Clinical Features of Non-Improved Patients

#### 3.4.1. Prognostic Factors for the Improvement of NASH

On the basis of the results above, we found that Ind-NASH was observed in both intraoperative and protocol liver biopsies. Figure 5 shows that while there were some patients with progression from NAFL or Ind-NASH to NASH, there were few cases with a progression of liver fibrosis. Therefore, we surmise that LSG has a metabolic effect on improving histopathological NASH findings.

Given this background, we investigated preoperative prognostic factors for improvements in NASH, via univariate and multivariate analyses. Univariate analyses revealed that HbA1c (7.1% vs. 8.4%; *p* = 0.0206), γ-GTP (54.7 IU/L vs. 88.1 IU/L; *p* = 0.0082), and type-4 collagen 7S (4.6 ng/mL vs. 5.8 ng/mL; *p* = 0.0263) were significant factors. We then undertook stepwise multivariate analyses, adding AST (52.3 IU/L vs. 88.1 IU/L; *p* = 0.0695) and the L/S ratio (0.9 vs. 0.7; *p* = 0.1464) (Table 3). Multivariate analyses revealed that the L/S ratio (odds ratio (OR): 0.002; 95% confidence interval (CI): <0.001–0.592; *p* = 0.0310) was the only prognostic factor of improvements in NASH (Table 3). However, the L/S ratio of the histopathological improvement group was lower than that of the non-improved group. To clarify the reasons underlying this result, we sequentially performed another analysis. The results are reported in the next section.

#### 3.4.2. Relationships between L/S Ratio and the Distribution of Ind-NASH at Intraoperative Liver Biopsy

The results of multivariate analyses derived an ROC curve of the L/S ratio that revealed a cut-off value (0.817) and AUC (0.657) for histopathological improvements in NASH at 12 months after LSG (Figure 9). Figure 9 also shows the distribution of each histopathological finding at intraoperative liver biopsy, with relationships among steatosis, the total NAFLD activity score, and the L/S ratio. The scatterplots indicated that Ind-NASH cases had low steatosis, low NAFLD activity scores, and higher L/S ratios.

#### 3.4.3. Clinical Features of Non-Improved Patients

We undertook univariate analyses using postoperative parameters in order to investigate the clinical features of patients who did not improve. This group showed a significant difference in %TWL (21.8% vs. 27.5%; *p* = 0.0331). Regarding glucose metabolic parameters, IRI (15.4 μU/mL vs. 8.5 μU/mL; *p* = 0.0122), HbA1c (6.1% vs. 5.5%; *p* = 0.0094), and HOMA-IR (4.2 vs. 1.8; *p* = 0.0153) were significant factors; therefore, patients with non-improved NASH still had stronger insulin resistance after LSG. With regard to NASH parameters, AST (27.7 IU/L vs. 16.5 IU/L; *p* = 0.0145), ALT (29.2 IU/L vs. 15.4 IU/L; *p* = 0.0004), MDA-LDL (123.2 U/L vs. 96.9 U/L; *p* = 0.0032), and liver volume (1921.7 mL vs. 1551.9 mL; *p* = 0.0071) were significant factors (Table 4). We undertook stepwise multivariate analyses using strongly significant parameters (*p* < 0.01); these multivariate analyses revealed that ALT (OR: 0.717; 95% CI: 0.178–0.953; *p* = 0.0081) was the only independent parameter remaining in NASH at 12 months after LSG (Table 4).

#### 3.4.4. Clinical Features of Non-Improve Patients

The ROC curve of ALT levels 12 months after LSG revealed a cut-off value (17.0 IU/L) and AUC (0.886) for histopathological improvements in NASH within that same timeframe (Figure 10). Scatterplots indicated that non-NASH cases 12 months after LSG had low steatosis and NAFLD activity scores, as well as lower ALT levels. These results present ALT as an independent prognostic factor of non-improved NASH at 12 months after LSG.

## 4. Discussion

In the present study, we found discrepancies between the current NASH diagnostic algorithm and actual histopathological findings. Such histopathological findings include some typical NASH findings, such as hepatocyte ballooning, inflammation, and fibrosis; however, they do not include steatosis (<5%). We term these findings “Ind-NASH”. Intraoperative and protocol liver biopsy revealed that Ind-NASH appeared in both cases, owing to preoperative and postoperative weight loss; in other words, preoperative diet and LSG reduced steatosis, but inflammation and/or fibrosis remained for a certain period. This finding highlights to a notable histopathological concept that has never been deeply discussed.

We clarified the clinical characteristics of patients with Ind-NASH. Patients with Ind-NASH were characterized by heavy body weight at the initial visit, but they showed good weight-loss effects after LSG. Liver dysfunction among Ind-NASH patients was much better than that observed in NASH patients, and the NAFIC score among Ind-NASH patients was the lowest, although initial metabolic disorders seemed to be worse in this subgroup, as insulin resistance was similar to that observed in NASH. However, histopathological findings were not made at the initial visit, but during LSG; therefore, the weight-loss effects derived from a very low calorie preoperative diet might improve NASH and convert it to Ind-NASH [18]. It is commonly believed that a minimum of 7% weight loss is needed for NASH resolution [19]. In our study population, the preoperative %TWL in patients with Ind-NASH was higher than in other groups (i.e., Ind-NASH vs. NASH vs. NAFL: 15.3% vs. 8.3% vs. 9.5%); however, our results could not demonstrate the previous results. If Ind-NASH represents only the result of a good preoperative weight-loss effect, we cannot explain the mechanism of a persistent weight-loss effect in patients with Ind-NASH; therefore, we must conclude that the clinical features of Ind-NASH differ from those of NASH/NAFL. Generally, people with sufficient pancreatic β-cell function can tolerate high calorie input and store it as visceral or ectopic fat deposition [5,20]; therefore, these patients tend to be heavier than those with normal pancreatic β-cell function. On the other hand, their metabolic parameters, such as FBG and TG, were lower than patients with NASH, as strong β-cell function controls these parameters until insulin secretion is exhausted. The preservation of pancreatic β-cell function results in effective preoperative and postoperative weight loss, as previously clarified.

This study made another novel scientific discovery. A protocol liver biopsy also revealed Ind-NASH as a condition between NASH and normal liver. Our protocol liver biopsy revealed an incremental improvement from NASH to normal liver via Ind-NASH in several patients. From these results, we determined that Ind-NASH was intrinsically NASH at the initial visit, but that steatosis was lost, owing to the weight-loss and metabolic effects of LSG. Our protocol liver biopsy revealed an incremental improvement from NASH to normal liver via Ind-NASH in several patients. On the basis of these results, we determined that Ind-NASH was intrinsically NASH at the initial visit; however, steatosis was lost due to the weight-loss and metabolic effects of LSG. In addition, some previous studies have demonstrated that liver steatosis improved more rapidly than live fibrosis after weight loss in patients with NASH [21,22]. However, no or minor steatosis (<5%) was observed in Ind-NASH. Therefore, Ind-NASH requires the eradication of liver steatosis, not only an improvement (≥5%). This assertion is supported by the fact that Ind-NASH does not lie between NAFL and normal liver, or between NAFL and NASH (Figure 11). To the best of our knowledge, the present study is the first to report on this phenomenon. The transition from NASH to Ind-NASH may occur from the preoperative phase, and we usually observe that transaminase first decreases as the inflammation improves. In addition, a preoperative very low calorie diet and potential hypersensitivity for weight-loss effects cause a dramatic reduction in hepatic fat accumulation [23]. Therefore, we conclude that improvements in inflammation and weight loss occur simultaneously.

Most practitioners may assume that patients with Ind-NASH have typical histopathological NASH on the initial visit as all patients are severely obese and have severe obesity-related diseases. However, we again emphasize that patients with Ind-NASH can achieve good weight loss, not only during the preoperative diet, but also during post-LSG periods. We encountered 33 Ind-NASH patients from their intraoperative liver biopsies to 12 months after LSGs. In these patients, postoperative %TWL was higher than in NASH-sustained patients (26.1% vs. 20.1%; *p* = 0.0331). In addition, FBG (92.3 mg/dL vs. 100.1 mg/dL; *p* = 0.0292), IRI (8.6 μU/mL vs. 14.6 μU/mL; *p* = 0.0058), and HOMA-IR (2.0 vs. 3.9; *p* = 0.0048) were significantly lower than those of NASH-sustained patients, regardless of whether they had T2D. Patients with Ind-NASH successfully achieved improved insulin resistance and pancreatic β-cell function.

We stratified the patients according to the histopathological findings of the intraoperative liver biopsy conducted in NASH, NAFL, and Ind-NASH groups. The true meaning of Ind-NASH should be clarified using paired liver biopsy both during the initial visit and during LSG. However, it should be noted that the cohort of the present study was limited to patients with severe obesity. In severely obese patients, performing a safe ultrasound-guided liver biopsy can be challenging due to thick subcutaneous fat. In addition, major complications, such as bleeding and biloma formation, should be avoided for conducting the present study whilst ensuring that it adheres to ethical regulations [24]. For these reasons, we chose intraoperative liver biopsy as the initial histopathological evaluation.

Eslam et al. established the concept of metabolic associated fatty liver disease (MAFLD) in 2020 [25]. According to this concept, presenting with overweight, obesity, or T2D with hepatic steatosis leads directly to an MAFLD diagnosis; therefore, morbidly obese patients are automatically diagnosed with MAFLD. The concept of MAFLD has rapidly spread, and practical guidelines for MAFLD were published in 2020 [26]. According to these guidelines, a liver biopsy should be considered when evidence of liver cirrhosis is detected [27]. However, is the timing of liver biopsies reasonable in patients undergoing MS? We previously reported that for morbidly obese patients, the negative predictive value of an NAFIC score and FIB-4 index was low (40.4%). There are some scoring systems for predicting advanced liver fibrosis, such as the NAFLD fibrosis score. However, morbidly obese Japanese patients frequently do not have advanced liver fibrosis; therefore, the positive predictive value was extremely low (8.3%) [5]. Given these findings—and due to a lack of sufficient clinical investigations—we need to interpret the too rapid infiltration of this concept carefully. At present, intraoperative liver biopsy is the most reasonable modality by which to diagnose NASH.

With regard to the diagnostic tools for NASH, the American Association for the Study of Liver Diseases published practical guidance in 2018 [27], wherein it recommends the NAFLD activity score and steatosis activity fibrosis as a scoring system [10,15]. Typical histopathological NASH consists of fatty degeneration, inflammatory cell infiltration, hepatocyte ballooning, Mallory–Denk body formation, and liver fibrosis. However, in the diagnostic process, the NAFLD activity score does not contain any fibrosis components. The FLIP algorithm is a tree diagram that is very useful in diagnosing NASH. However, the FLIP algorithm is not indicated when steatosis is less than 5%; therefore, it defines Ind-NASH as non-NAFLD. These deviations in diagnostic tools might induce the concept of Ind-NASH as a discrepancy.

Our protocol liver biopsy revealed the significant histopathological resolution of NASH by LSG. Upon including NAFL/Ind-NASH patients, the mean steatosis percentage decreased from 25.2% to 8.0%. Furthermore, the mean total NAFLD activity score decreased from 2.7 to 1.2, and the Matteoni type also decreased from 2.7 to 1.3, even 12 months after LSG. In this study, 15 patients (42.9%) achieved histopathological NASH remission until 12 months after LSG; however, neither the PFS nor the Brunt staging scores significantly improved [5]. Several previous studies of morbidly obese patients have confirmed the effect of LSG on NASH and liver fibrosis; however, some studies could not demonstrate improvements in liver fibrosis by bariatric procedures. Lassailly et al. recently reported a long-term cohort study with confirmed liver histology, clarifying that NASH resolved in 84% of cases and that median fibrosis also significantly decreased (2.0 vs. 1.0; *p* < 0.001) at one year after bariatric surgery [28,29]. They also clarified that the resolution of liver fibrosis continued for five years after bariatric surgery [29]. Therefore, the resolution of liver fibrosis requires a long-term study relative to steatosis reduction and inflammation improvement.

We previously clarified that the prevalence of T2D and type-4 collagen 7S are independent prognostic factors of liver fibrosis [9]. In the present study, the initial L/S ratio was the only prognostic factor of histopathological improvement 12 months after LSG. Scatterplots also revealed that Ind-NASH exhibited low steatosis and low total NAFLD activity scores. Our previous study revealed that the mean L/S ratio improved from 0.81 to 1.03 by virtue of a very low calorie preoperative diet, and that the preoperative L/S ratio closely correlated with the steatosis percentage of intraoperative liver biopsy (correlation coefficient = −0.749; *p* < 0.001) [5]. The very low calorie preoperative diet also contributed to the preoperative liver volume reduction (5–20%; mean: 14%) [18]. Therefore, patients with Ind-NASH may be good candidates for MS, owing to their good treatment response. Currently, various non-invasive tests for the diagnosis of NASH have been reported [30,31,32], and the AUCs of these scoring systems are quite accurate [13,30]. However, we demonstrated that preoperative scoring systems for NASH do not correctly reflect histopathological findings when applied to severely obese patients [5].

By contrast, ALT levels 12 months after LSG constituted independent clinical parameters of NASH residue. There are corresponding physiological constraints: The degree of liver enzyme elevation, for example, correlates with the severity or sustainment of hepatocellular injury [33]. Changes in liver enzymes also correlate with a reduction in BMI with an improvement in metabolic function, reducing both insulin resistance and oxidative stress [34,35,36]. Our univariate analyses revealed that insulin resistance (IRI, HbA1c, and HOMA-IR) and oxidative stress (MDA-LDL) were significant factors [37]. Most non-invasive tests for the diagnosis of NAFLD include AST/ALT levels [32]; however, Verma et al. reported that the AUCs for ALT levels correlating with NASH and advanced fibrosis were 0.62 and 0.46, respectively [38]. The AUC of ALT was very high (0.866), and the cut-off value was very strict; however, we could clearly stratify patients in terms of histopathological improvements in NASH.

This study has several limitations that must be acknowledged. First, MS is not widely used in Japan, the number of subjects herein is relatively small, and the study is from a single institution and retrospective in nature. Second, the follow-up period was short; before there can be a definitive conclusion on this issue, long-term studies that include many patients are needed in the future. Third, we could not enroll patients who had undergone a malabsorptive procedure, such as Roux-en Y gastric bypass (RYGB) or duodenojejunal bypass with sleeve gastrectomy, as LSG is the only MS procedure covered by Japan’s national health insurance system. However, previous studies have revealed that RYGB is superior to LSG in resolving steatosis, even though there is no significant difference between these two procedures in improving liver fibrosis one year after surgery [39,40,41]. Therefore, a long-term histopathological evaluation is warranted to confirm improvements in liver fibrosis in these patients. Regarding the accuracy of liver biopsy, it has been shown to have a high rate of sampling error as the liver may have both focal and whole histopathological changes due to its volume. To supplement this uncertainty, FibroScan and magnetic resonance imaging may be useful modalities to combine with liver biopsy [42,43,44]. Nonetheless, we did not apply these modalities in the present study. Therefore, future studies employing a combination of these modalities are warranted.

## 5. Conclusions

By evaluating the protocol liver biopsy from the intraoperative period to 12 months after LSG, we established a brand-new concept of indeterminable nonalcoholic steatohepatitis (Ind-NASH). The important histopathological indication of Ind-NASH is fibrosis and/or inflammation residue without steatosis, a histopathological finding that has not been clearly defined. Ind-NASH was observed in this study as an intermediate condition between NASH and normal liver; therefore, Ind-NASH may represent a recovery process from NASH by MS. The clinical features of patients with Ind-NASH were as follows: initial body weight was higher than that of patients with NASH, insulin resistance was relatively strong, and liver dysfunction was not worse than that of NASH patients. Patients with Ind-NASH may potentially observe a good weight-loss effect from a preoperative diet to the postoperative course.

In conclusion, patients with Ind-NASH may be good candidates for MS, owing to their good treatment response. For these reasons, Ind-NASH should be deeply discussed and clearly defined by specialists in the near future.

## Figures and Tables

**Figure 1 biomedicines-10-00453-f001:**
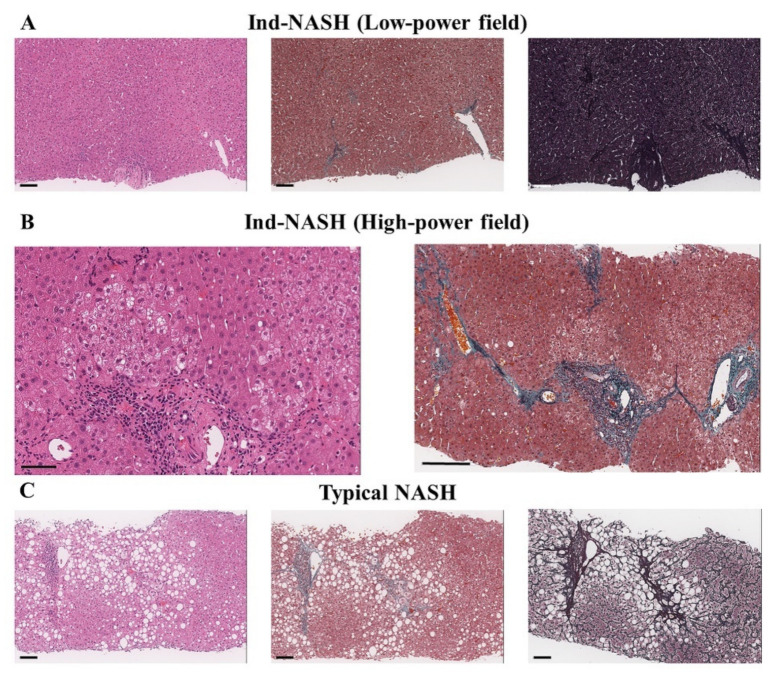
Histopathological differences between Ind-NASH and typical NASH. Microscopic findings are shown as hematoxylin–eosin, Masson trichrome, and silver reticulin from the left to right. (**A**) A typical Ind-NASH patient with a low-power field view. Small bars represent 100 μm. Mild steatosis and mild abnormal fibrosis around central veins are seen. (**B**) Focal ballooning in an Ind-NASH patient (left). Small bar represents 60 μm. Central vein fibrosis without steatosis (right). Small bar represents 200 μm. (**C**) A typical NASH patient. Small bars represent 100 μm. Severe steatosis, hepatocyte ballooning, and periportal fibrosis are seen.

**Figure 2 biomedicines-10-00453-f002:**
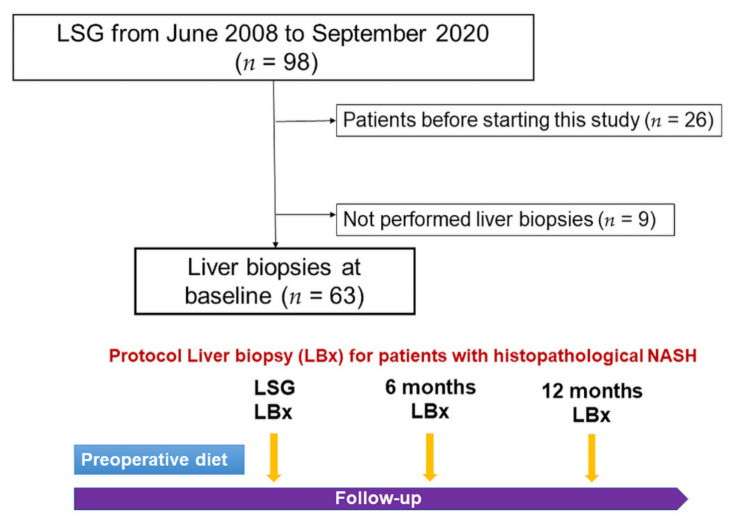
A flowchart of the eligible patients included in this study. Protocol intraoperative liver biopsy was performed on patients with histopathological NASH. Abbreviation: LB, liver biopsy.

**Figure 3 biomedicines-10-00453-f003:**
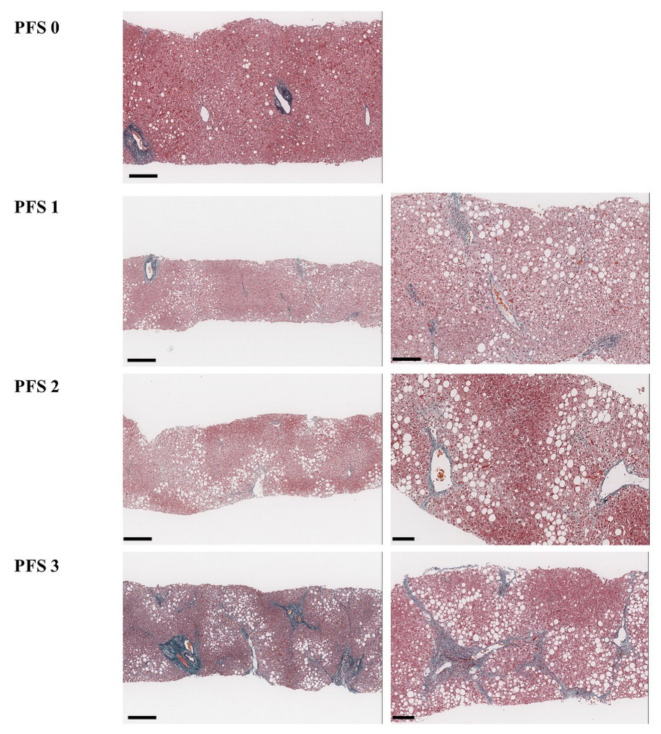
Histopathological definitions of PFS. PFS 0, no liver fibrosis, and the small bars represent 200 μm. PFS 1, pericellular fibrosis confined to the proximity of central veins and is present in approximately <50% of these veins; high-power field view (right). Small bars represent 300 μm and 100 μm (from the left). PFS 2, pericellular fibrosis confined to the proximity of central veins and is present in > 50% of the central veins; high-power field view (right). Small bars represent 200 μm and 100 μm (from the left). PFS 3, pericellular fibrosis around the central veins with periportal fibrosis or bridging fibrosis; high-power field view (right). Small bars represent 200 μm and 100 μm (from the left).

**Figure 4 biomedicines-10-00453-f004:**
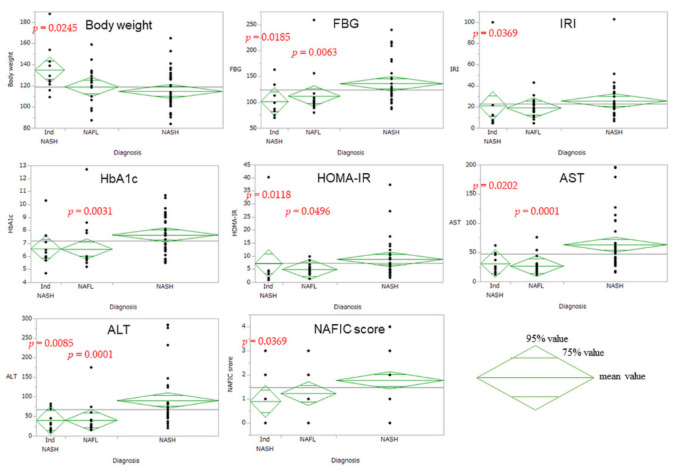
Upper and lower vertical angles of rhombuses indicate the 95% value. The control arm is the NASH group; therefore, if the vertical angles are not overlapping, it means significant differences compared to the NASH group.

**Figure 5 biomedicines-10-00453-f005:**
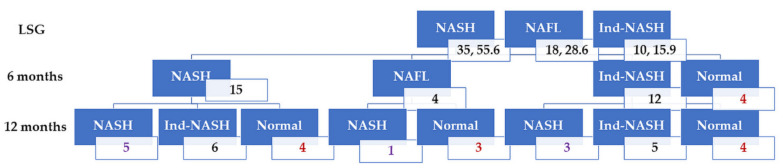
Patient flowcharts are shown. At 6 months after LSG, only 35 patients with NASH received liver biopsy. Four patients were able to achieve histopathological remission at 6 months after LSG. Therefore, 31 patients received liver biopsy at 12 months after LSG.

**Figure 6 biomedicines-10-00453-f006:**
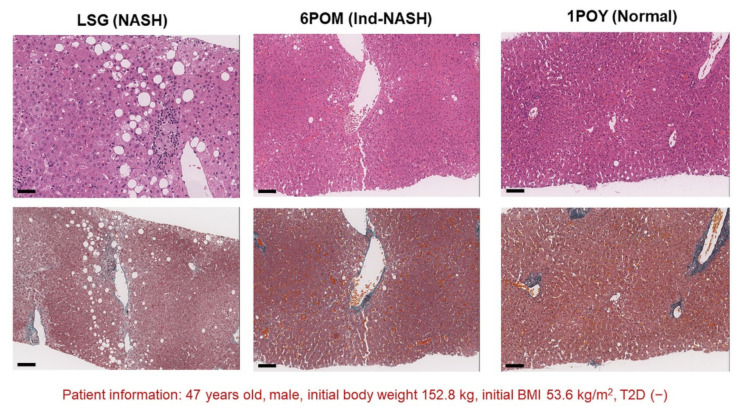
Protocol live biopsy confirmed sequential histopathological improvement in the patient with good weight-loss effects before and after LSG. BMI at LSG, 6 months after LSG (6POM), and 12 months after LSG (12POM) were 47.2 kg/m^2^, 43.6 kg/m^2^, and 40.0 kg/m^2^, respectively. The patient was able to achieve histopathological remission of NASH at 12 months after LSG. Small bars represent 50 μm. Abbreviations: 6POM, 6 months after LSG; 12POM, 12 months after LSG.

**Figure 7 biomedicines-10-00453-f007:**
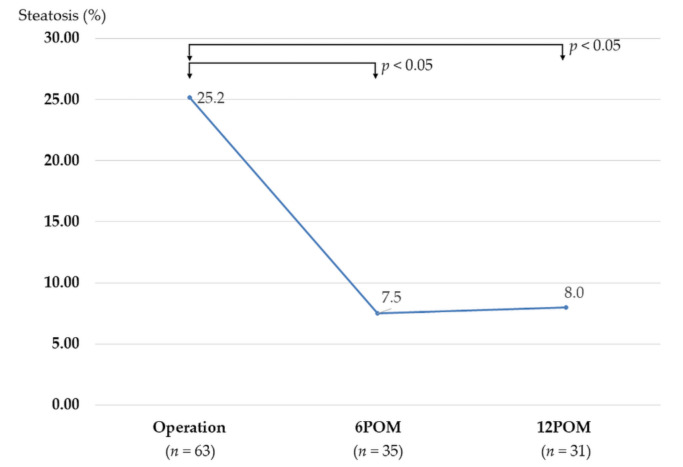
Changes in the percentages of steatosis evaluated by protocol liver biopsy. Abbreviations: 6POM, 6 months after LSG; 12POM, 12 months after LSG.

**Figure 8 biomedicines-10-00453-f008:**
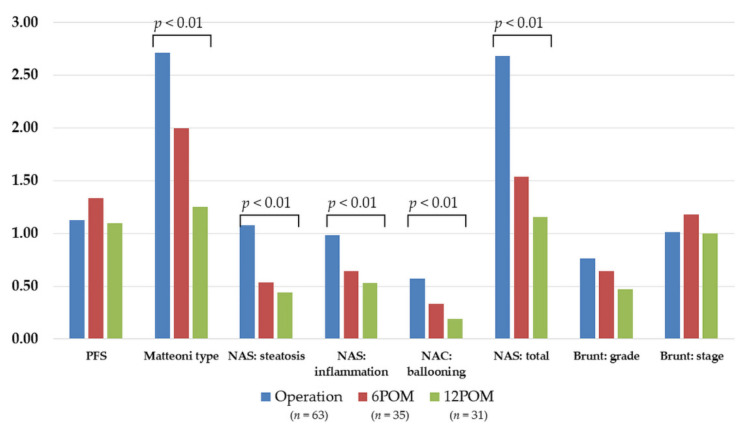
Changes in histopathological scoring and classifications of NASH evaluated by protocol liver biopsy. Abbreviations: PFS, Pericellular fibrosis score; NAS, NAFLD activity score; 6POM, 6 months after LSG; 12POM, 12 months after LSG.

**Figure 9 biomedicines-10-00453-f009:**
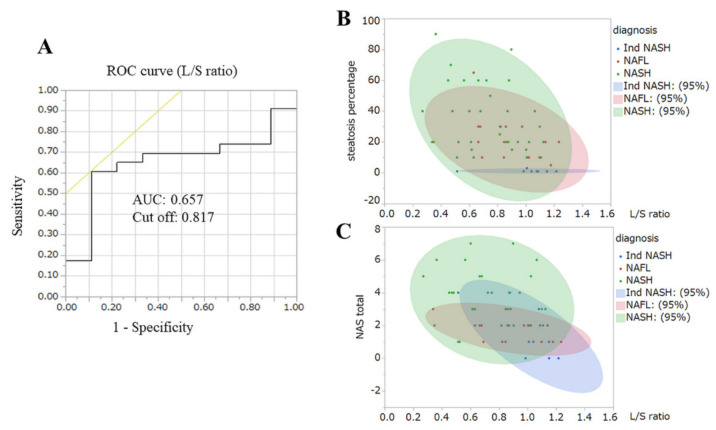
(**A**) The ROC curve revealed the cut-off and AUC of L/S ratio. (**B**) Scatter plots between steatosis and L/S ratio. Almost all blue plots (Ind-NASH) were in higher L/S ratio locations. (**C**) Almost all blue plots (Ind-NASH) were in low NAFLD activity score locations.

**Figure 10 biomedicines-10-00453-f010:**
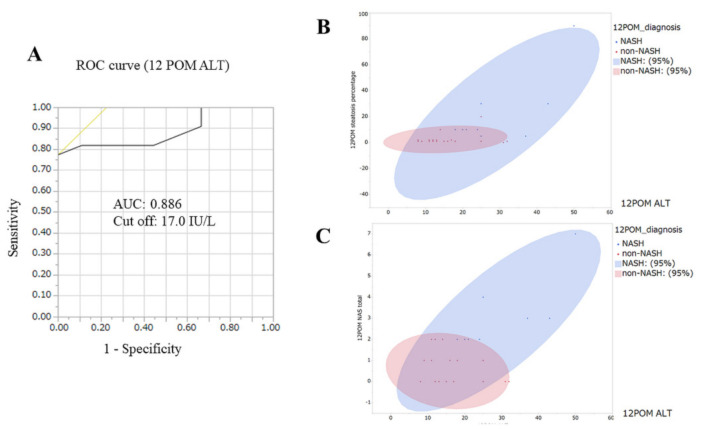
(**A**) The ROC curve revealed the cut-off and AUC of ALT. (**B**) Scatter plots between steatosis and L/S ratio. Almost all red plots (non-NASH) were in both lower ALT level and lower steatosis percentage locations at 12 months after LSG. (**C**) Almost all red plots (non-NASH) were in low NAFLD activity score locations with lower ALT levels.

**Figure 11 biomedicines-10-00453-f011:**
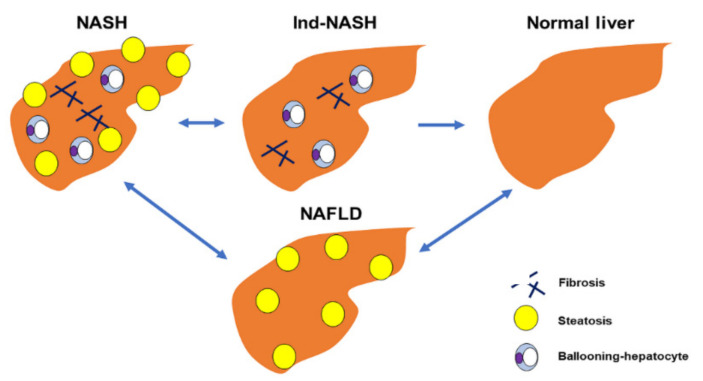
The brand-new concept of Ind-NASH is shown. Ind-NASH lies between NASH and normal liver but does not lie between NAFL and NASH.

**Table 1 biomedicines-10-00453-t001:** Baseline characteristics of enrolled patients and stratifications along with intraoperative liver biopsy.

	All Patients (*n* = 63)	NASH (*n* = 35)	NAFL (*n* = 18)	Ind-NASH (*n* = 10)	*p*-Value (Control: NASH)
Age, years	43.3 ± 12.3	41.8 ± 13.9	44.4 ± 8.2	46.4 ± 13.0	NAFL: 0.5814 Ind-NASH: 0.6729
Male, *n* (%)	32 (50.8)	15 (42.9)	9 (50.0)	8 (80.0)	NAFL: 0.8598 Ind-NASH: 0.0807
Body weight, kg	119.1 ± 20.5	114.7 ± 18.8	119.0 ± 18.9	134.9 ± 22.9	NAFL: 0.6352 * Ind-NASH: 0.0245
BMI, kg/m^2^	43.6 ± 6.3	42.6 ± 5.2	43.1 ± 5.6	48.1 ± 9.1	NAFL: 0.9670 Ind-NASH: 0.1237
Comorbidities, *n* HT, *n* (%) T2D, *n* (%) HL, *n* (%) HU, *n* (%) OSA, *n* (%)	5.7 ± 2.2 53 (84.1) 40 (63.5) 45 (71.4) 30 (47.6) 60 (95.2)	5.7 ± 2.1 28 (80.0) 28 (80.0) 25 (71.4) 12 (34.3) 33 (94.3)	5.4 ± 2.4 15 (83.3) 6 (33.3) 13 (72.2) 10 (55.6) 17 (94.4)	6.2 ± 2.3 10 (100.0) 6 (60.0) 7 (70.0) 8 (80.0) 10 (100.0)	
FBG, mg/dL	123.9 ± 42.1	135.8 ± 41.9	112.1 ± 41.2	101.2 ± 30.5	* NAFL: 0.0063 * Ind-NASH: 0.0185
IRI, μU/mL	23.1 ± 18.8	25.6 ± 18.2	19.0 ± 9.4	21.1 ± 32.2	NAFL: 0.4055 * Ind-NASH: 0.0369
HbA1c, %	7.2 ± 1.7	7.7 ± 1.5	6.6 ± 1.9	6.6 ± 1.5	* NAFL: 0.0031 Ind-NASH: 0.0619
HOMA-IR, no unit	7.4 ± 7.6	8.7 ± 7.4	4.9 ± 2.1	7.1 ± 13.5	* NAFL: 0.0496 * Ind-NASH: 0.0188
HOMA-β, no unit	172.7 ± 121.5	162.5 ± 119.4	197.7 ± 123.1	162.5 ± 135.6	NAFL: 0.3235 Ind-NASH: 0.9762
C-peptide, ng/mL	2.9 ± 1.3	3.3 ± 1.4	2.7 ± 0.9	2.2 ± 1.0	NAFL: 0.3366 Ind-NASH: 0.0952
TC, mg/dL	187.6 ± 36.5	185.4 ± 36.9	191.4 ± 40.4	188.2 ± 30.1	NAFL: 0.8699 Ind-NASH: 0.9980
TG, mg/dL	138.9 ± 81.2	147.3 ± 94.3	128.2 ± 68.2	128.8 ± 49.1	NAFL: 0.6288 Ind-NASH: 0.9921
LDL-C, mg/dL	120.0 ± 30.9	114.9 ± 30.0	127.2 ± 34.8	124.7 ± 25.1	NAFL: 0.4969 Ind-NASH: 0.5634
HDL-C, mg/dL	43.5 ± 10.3	45.2 ± 11.9	42.9 ± 7.6	38.7 ± 6.7	NAFL: 0.8597 Ind-NASH: 0.2546
AST, IU/L	47.7 ± 40.8	63.2 ± 47.7	26.8 ± 16.3	31.0 ± 16.7	* NAFL: 0.0001 * Ind-NASH: 0.0202
ALT, IU/L	67.9 ± 62.3	90.4 ± 70.7	39.9 ± 37.7	39.4 ± 27.2	* NAFL: 0.0001 * Ind-NASH: 0.0085
γ-GTP, IU/L	60.0 ± 54.0	75.0 ± 65.8	42.3 ± 21.2	39.5 ± 30.4	NAFL: 0.2592 Ind-NASH: 0.1429
Albumin, g/dL	4.3 ± 0.4	4.3 ± 0.4	4.3 ± 0.4	4.2 ± 0.5	NAFL: 0.7084 Ind-NASH: 0.7871
T-Bil, mg/dL	0.8 ± 0.4	0.8 ± 0.3	0.7 ± 0.4	1.0 ± 0.6	NAFL: 0.3930 Ind-NASH: 0.7521
Ferritin, ng/mL	154.8 ± 129.8	171.2 ± 145.7	127.6 ± 84.2	140.1 ± 133.6	NAFL: 0.7902 Ind-NASH: 0.7825
Type-4 collagen, ng/mL	4.8 ± 1.2	5.1 ± 1.3	4.5 ± 0.8	4.4 ± 1.2	NAFL: 0.1940 Ind-NASH: 0.2673
Hyaluronic acid, ng/mL	31.7 ± 25.7	35.3 ± 30.8	27.1 ± 13.4	25.9 ± 19.5	NAFL: 0.9997 Ind-NASH: 0.4679
MDA-LDL, U/L	140.4 ± 56.9	144.3 ± 54.4	123.9 ± 57.1	164.3 ± 66.9	NAFL: 0.4414 Ind-NASH: 0.8297
Transferrin, mg/dL	270.0 ± 43.9	273.1 ± 42.9	264.3 ± 43.0	268.2 ± 53.2	NAFL: 0.6456 Ind-NASH: 0.9510
PAI-1, ng/mL	53.7 ± 38.0	60.5 ± 36.5	39.8 ± 16.0	51.9 ± 62.4	NAFL: 0.0982 Ind-NASH: 0.1388
FIB-4 index, no unit	0.9 ± 0.7	1.0 ± 0.9	0.7 ± 0.4	1.0 ± 0.4	NAFL: 0.7834 Ind-NASH: 0.7107
NAFIC score, no unit	1.5 ± 1.1	1.8 ± 1.1	1.2 ± 0.8	0.9 ± 1.3	NAFL: 0.1345 * Ind-NASH: 0.0369
NAFLD fibrosis score, no unit	0.1 ± 1.7	0.1 ± 1.9	0.8 ± 1.1	0.2 ± 1.4	NAFL: 0.0595 Ind-NASH: 0.9989
SFA, cm^2^	524.7 ± 133.5	515.9 ± 142.6	529.0 ± 132.6	554.2 ± 98.9	NAFL: 0.9305 Ind-NASH: 0.5172
VFA, cm^2^	272.8 ± 81.0	269.6 ± 81.4	267.2 ± 77.5	298.5 ± 92.3	NAFL: 0.9140 Ind-NASH: 0.5796
Waist, cm	122.0 ± 9.5	121.0 ± 10.4	121.8 ± 9.2	126.6 ± 4.4	NAFL: 0.9910 Ind-NASH: 0.1126
Liver volume, mL	2249.0 ± 542.7	2250.9 ± 578.6	2225.7 ± 443.1	2286.1 ± 625.2	NAFL: 0.9949 Ind-NASH: 1.0000
L/S ratio, no unit	0.8 ± 0.2	0.8 ± 0.2	0.9 ± 0.2	1.0 ± 0.2	NAFL: 0.2985 * Ind-NASH: 0.0037

Values are the mean ± standard deviation. * Parameters with *p* < 0.05. Abbreviations: BMI, body mass index; HT, hypertension; T2D, type 2 diabetes; HL, hyperlipidemia; HU, hyperuricemia; OSA, obstructive sleep apnea; FBG, fasting blood glucose; IRI, immunoreactive insulin; HbA1c, hemoglobin A1c; HOMA-IR, homeostasis model for assessing insulin resistance; HOMA-β, homeostasis model assessment beta cell function; TC, total cholesterol; TG, triglyceride; LDL-C, low-density lipoprotein cholesterol; HDL-C, high-density lipoprotein cholesterol; AST, aspartate aminotransferase; ALT, alanine aminotransferase; γ-GTP, γ-glutamyl transpeptidase; T-Bil, total bilirubin; BUN, blood urea nitrogen; eGFR, estimated glomerular filtration rate; MDA-LDL, malondialdehyde-modified low-density lipoprotein cholesterol; PAI-1, plasminogen activator inhibitor 1; SFA, subcutaneous fat area; VFA, visceral fat area; L/S ratio, liver/spleen ratio; NASH, nonalcoholic steatohepatitis; NAFL, nonalcoholic fatty liver; Ind-NASH, indeterminable NASH.

**Table 2 biomedicines-10-00453-t002:** Weight-loss effects and changes in metabolic parameters before and after LSG.

	Initial	6 Months after LSG	12 Months after LSG	*p*-Value Initial vs. 12 Months
**All Patients (*n* = 63)**				
Body weight, kg	119.1 ± 20.5	89.4 ± 14.1	87.6 ± 15.7	<0.0001
BMI, kg/m^2^	43.6 ± 6.3	32.7 ± 4.0	31.9 ± 4.6	<0.0001
%TWL, %	-	24.5 ± 6.5	26.7 ± 7.5	-
FBG, mg/dL	123.9 ± 42.1	91.0 ± 13.6	92.5 ± 16.5	<0.0001
IRI, μU/mL	23.1 ± 18.8	8.1 ± 4.8	8.6 ± 6.1	<0.0001
HbA1c, %	7.2 ± 1.7	5.7 ± 0.7	5.6 ± 0.7	<0.0001
HOMA-IR, no unit	7.4 ± 7.6	1.8 ± 1.2	2.0 ± 1.9	<0.0001
HOMA-β, no unit	172.7 ± 121.5	113.0 ± 89.3	115.9 ± 80.8	0.0053
AST, IU/L	47.7 ± 40.8	19.7 ± 13.3	18.5 ± 9.6	<0.0001
ALT, IU/L	67.9 ± 62.3	22.5 ± 32.5	17.9 ± 8.8	<0.0001
γ-GTP, IU/L	60.0 ± 54.0	31.1 ± 61.1	34.3 ± 69.3	0.0262
LDL-C, mg/dL	120.0 ± 30.9	111.4 ± 28.2	108.7 ± 30.1	0.0479
HDL-C, mg/dL	43.5 ± 10.3	52.5 ± 10.9	58.2 ± 13.1	<0.0001
TG, mg/dL	138.9 ± 81.2	94.8 ± 52.5	90.3 ± 56.0	0.0002
Ferritin, ng/mL	154.8 ± 129.8	93.2 ± 70.6	84.4 ± 71.7	0.0006
Type-4 collagen 7S, ng/mL	4.8 ± 1.2	4.7 ± 0.9	6.1 ± 11.9	0.4225
Hyaluronic acid, ng/mL	31.7 ± 25.7	34.9 ± 28.4	37.3 ± 30.7	0.2973
MDA-LDL, U/L	140.4 ± 56.9	111.1 ± 40.5	106.9 ± 35.6	0.0004
Transferrin, mg/dL	270.0 ± 43.9	224.7 ± 46.7	257.6 ± 50.5	0.1621
PAI-1, ng/mL	53.7 ± 38.0	21.6 ± 15.1	20.0 ± 13.3	<0.0001
SFA, cm^2^	524.7 ± 133.5	339.2 ± 109.9	340.8 ± 133.4	<0.0001
VFA, cm^2^	272.8 ± 81.0	161.4 ± 70.0	145.6 ± 67.2	<0.0001
Waist, cm	122.0 ± 9.5	104.8 ± 11.7	103.0 ± 13.0	<0.0001
Liver volume, mL	2249.0 ± 542.7	1674.2 ± 332.0	1654.9 ± 316.2	<0.0001
L/S ratio, no unit	0.8 ± 0.2	1.3 ± 0.2	1.3 ± 0.2	<0.0001
**NASH (*n* = 35)**				
Body weight, kg	114.7 ± 18.8	87.1 ± 13.5	86.2 ± 15.3	<0.0001
BMI, kg/m^2^	42.6 ± 5.2	32.2 ± 3.4	31.7 ± 4.0	<0.0001
%TWL, %	-	23.7 ± 5.9	25.7 ± 6.7	-
FBG, mg/dL	135.8 ± 41.9	91.7 ± 17.9	94.7 ± 17.9	<0.0001
IRI, μU/mL	25.6 ± 18.2	10.2 ± 5.0	10.5 ± 6.7	<0.0001
HbA1c, %	7.7 ± 1.5	5.7 ± 0.6	5.8 ± 0.7	<0.0001
HOMA-IR, no unit	8.7 ± 7.4	2.3 ± 1.3	2.7 ± 2.3	0.0001
HOMA-β, no unit	162.5 ± 119.4	138.9 ± 82.1	125.1 ± 61.9	0.1342
AST, IU/L	63.2 ± 47.7	21.7 ± 16.7	19.6 ± 12.0	<0.0001
ALT, IU/L	90.4 ± 70.7	27.8 ± 42.6	18.7 ± 10.1	<0.0001
γ-GTP, IU/L	75.0 ± 65.8	38.5 ± 80.4	45.5 ± 92.3	0.1456
LDL-C, mg/dL	114.9 ± 30.0	109.8 ± 27.7	102.4 ± 30.3	0.1038
HDL-C, mg/dL	45.2 ± 11.9	53.1 ± 12.2	59.3 ± 14.7	<0.0001
TG, mg/dL	147.3 ± 94.3	104.5 ± 59.8	100.4 ± 68.3	0.0236
Ferritin, ng/mL	171.2 ± 145.7	86.5 ± 71.6	77.5 ± 68.9	0.0017
Type-4 collagen, ng/mL	5.1 ± 1.3	4.7 ± 1.0	7.4 ± 16.3	0.4578
Hyaluronic acid, ng/mL	35.3 ± 30.8	31.6 ± 28.9	33.3 ± 27.1	0.7775
MDA-LDL, U/L	144.3 ± 54.4	110.6 ± 31.9	103.5 ± 26.3	0.0005
Transferrin, mg/dL	273.1 ± 42.9	250.9 ± 43.3	264.2 ± 47.9	0.4357
PAI-1, ng/mL	60.5 ± 36.5	24.5 ± 18.2	23.5 ± 16.4	<0.0001
SFA, cm^2^	515.9 ± 142.6	337.3 ± 103.8	347.9 ± 117.5	<0.0001
VFA, cm^2^	269.6 ± 81.4	156.8 ± 63.3	152.3 ± 71.7	<0.0001
Waist, cm	121.0 ± 10.4	103.8 ± 11.7	103.2 ± 12.6	<0.0001
Liver volume, mL	2250.9 ± 578.6	1637.9 ± 328.9	1618.5 ± 347.8	<0.0001
L/S ratio, no unit	0.8 ± 0.2	1.3 ± 0.3	1.3 ± 0.3	<0.0001
**NAFL (*n* = 18)**				
Body weight, kg	119.0 ± 18.9	89.8 ± 13.7	88.0 ± 15.6	<0.0001
BMI, kg/m^2^	43.1 ± 5.6	32.5 ± 4.2	31.9 ± 5.0	<0.0001
%TWL, %	-	24.2 ± 6.4	25.9 ± 7.3	-
FBG, mg/dL	112.1 ± 41.2	91.0 ± 17.2	87.3 ± 12.6	0.0283
IRI, μU/mL	19.0 ± 9.4	5.5 ± 2.4	6.9 ± 4.8	0.0001
HbA1c, %	6.6 ± 1.9	5.6 ± 0.7	5.5 ± 0.7	0.0342
HOMA-IR, no unit	4.9 ± 2.1	1.2 ± 0.7	1.4 ± 0.9	<0.0001
HOMA-β, no unit	197.7 ± 123.1	62.7 ± 42.8	121.6 ± 106.4	0.0751
AST, IU/L	26.8 ± 16.3	16.8 ± 7.4	15.9 ± 4.3	0.0137
ALT, IU/L	39.9 ± 37.7	14.9 ± 7.6	15.1 ± 4.9	0.0124
γ-GTP, IU/L	42.3 ± 21.2	21.3 ± 17.5	18.1 ± 8.5	0.0002
LDL-C, mg/dL	127.2 ± 34.8	113.7 ± 27.6	115.4 ± 21.6	0.2349
HDL-C, mg/dL	42.9 ± 7.6	52.7 ± 8.0	56.9 ± 9.6	<0.0001
TG, mg/dL	128.2 ± 68.2	82.5 ± 39.9	76.6 ± 27.0	0.0071
Ferritin, ng/mL	127.6 ± 84.2	91.4 ± 67.1	88.9 ± 84.8	0.2121
Type-4 collagen, ng/mL	4.5 ± 0.8	4.7 ± 0.8	4.5 ± 0.8	0.9823
Hyaluronic acid, ng/mL	27.1 ± 13.4	39.4 ± 27.4	39.1 ± 23.1	0.0869
MDA-LDL, U/L	123.9 ± 57.1	102.2 ± 41.7	99.0 ± 26.5	0.1277
Transferrin, mg/dL	264.3 ± 43.0	242.7 ± 51.6	255.5 ± 60.1	0.6394
PAI-1, ng/mL	39.8 ± 16.0	20.4 ± 10.8	16.1 ± 6.1	<0.0001
SFA, cm^2^	529.0 ± 132.6	340.2 ± 130.3	339.7 ± 160.6	0.0009
VFA, cm^2^	267.2 ± 77.5	170.1 ± 85.5	143.9 ± 64.6	<0.0001
Waist, cm	121.8 ± 9.2	104.8 ± 12.4	103.2 ± 14.6	0.0002
Liver volume, mL	2225.7 ± 443.1	1645.4 ± 307.8	1633.5 ± 236.2	<0.0001
L/S ratio, no unit	0.9 ± 0.2	1.3 ± 0.1	1.3 ± 0.2	<0.0001
**Ind-NASH (*n* = 10)**				
Body weight, kg	134.9 ± 22.9	96.8 ± 15.4	92.2 ± 18.1	0.0003
BMI, kg/m^2^	48.1 ± 9.1	34.4 ± 5.5	32.9 ± 6.1	0.0006
%TWL, %	-	27.8 ± 7.8	32.0 ± 9.4	-
FBG, mg/dL	101.2 ± 30.5	88.8 ± 14.9	94.9 ± 17.7	0.5989
IRI, μU/mL	21.1 ± 32.2	4.8 ± 2.9	5.0 ± 3.5	0.2044
HbA1c, %	6.6 ± 1.5	5.5 ± 0.7	5.5 ± 0.7	0.0638
HOMA-IR, no unit	7.1 ± 13.5	1.1 ± 0.6	1.2 ± 0.8	0.2527
HOMA-β, no unit	162.5 ± 135.6	107.7 ± 133.7	79.2 ± 81.6	0.1590
AST, IU/L	31.0 ± 16.7	17.6 ± 4.6	19.6 ± 6.8	0.0692
ALT, IU/L	39.4 ± 27.2	17.3 ± 5.7	20.7 ± 9.5	0.0655
γ-GTP, IU/L	39.5 ± 30.4	22.4 ± 13.8	26.0 ± 19.3	0.2607
LDL-C, mg/dL	124.7 ± 25.1	113.2 ± 33.2	116.6 ± 40.9	0.6145
HDL-C, mg/dL	38.7 ± 6.7	50.3 ± 11.3	56.7 ± 13.9	0.0047
TG, mg/dL	128.8 ± 49.1	83.0 ± 40.6	82.8 ± 48.2	0.0551
Ferritin, ng/mL	140.1 ± 133.6	119.2 ± 74.5	100.8 ± 60.1	0.4377
Type-4 collagen, ng/mL	4.4 ± 1.2	5.0 ± 0.7	5.1 ± 0.9	0.1827
Hyaluronic acid, ng/mL	25.9 ± 19.5	38.7 ± 29.3	47.4 ± 50.1	0.2567
MDA-LDL, U/L	164.3 ± 66.9	128.9 ± 57.6	131.6 ± 61.7	0.3605
Transferrin, mg/dL	268.2 ± 53.2	227.7 ± 48.5	239.0 ± 39.7	0.1034
PAI-1, ng/mL	51.9 ± 62.4	14.0 ± 4.5	15.1 ± 5.7	0.1161
SFA, cm^2^	554.2 ± 98.9	343.8 ± 100.5	320.5 ± 141.5	0.0014
VFA, cm^2^	298.5 ± 92.3	160.6 ± 66.3	127.6 ± 60.2	0.0008
Waist, cm	126.6 ± 4.4	108.1 ± 11.3	101.8 ± 13.1	0.0007
Liver volume, mL	2286.1 ± 625.2	1846.0 ± 363.0	1805.9 ± 325.7	0.0872
L/S ratio, no unit	1.0 ± 0.2	1.2 ± 0.1	1.3 ± 0.3	0.0109

Values are the mean ± standard deviation. Abbreviations: LSG, laparoscopic sleeve gastrectomy; BMI, body mass index; %TWL, percentage total weight loss; FBG, fasting blood glucose; IRI, immunoreactive insulin; HbA1c, hemoglobin A1c; HOMA-IR, homeostasis model for assessing insulin resistance; HOMA-β, homeostasis model assessment beta cell function; AST, aspartate aminotransferase; ALT, alanine aminotransferase; γ-GTP, γ-glutamyl transpeptidase; TG, triglyceride; LDL-C, low-density lipoprotein cholesterol; HDL-C, high-density lipoprotein cholesterol; MDA-LDL, malondialdehyde-modified low density lipoprotein cholesterol; PAI-1, plasminogen activator inhibitor 1; SFA, subcutaneous fat area; VFA, visceral fat area; LS ratio, liver/spleen ratio; NASH, nonalcoholic steatohepatitis; NAFL, nonalcoholic fatty liver; Ind-NASH, indeterminable NASH.

**Table 3 biomedicines-10-00453-t003:** Univariate and multivariate analyses of prognostic factors for improvement of NASH at 12 months after LSG.

Parameters	Improvement (*n* = 23)	Non-Improvement (*n* = 9)		Odds Ratio (95% Confidence Interval)	*p*-Value
**Preoperative Parameters**					
Male, *n* (%)	12 (52.2%)	6 (66.7%)		1.833 (0.382–10.404)	0.4541
BMI, kg/m^2^	44.1 ± 5.5	42.0 ± 5.1		1.083 (0.933–1.286)	0.3014
FBG, mg/dL	127.5 ± 45.4	147.8 ± 33.1		0.989 (0.970–1.007)	0.2330
IRI, μU/mL	26.8 ± 20.7	37.1 ± 29.2		0.982 (0.945–1.017)	0.3037
HbA1c, %	7.1 ± 1.5	8.4 ± 1.3		0.532 (0.279–0.909)	0.0206 *
HOMA-IR, no unit	8.8 ± 9.1	13.9 ± 10.9		0.950 (0.865–1.032)	0.2215
AST, IU/L	52.3 ± 38.4	88.1 ± 64.1		0.986 (0.968–1.001)	0.0695
ALT, IU/L	81.4 ± 60.6	115.2 ± 96.5		0.994 (0.983–1.004)	0.2471
γ-GTP, IU/L	54.7 ± 47.5	126.6 ± 88.1		0.984 (0.968–0.996)	0.0082 *
Type-4 collagen 7S, ng/mL	4.6 ± 1.2	5.8 ± 1.5		0.508 (0.235–0.968)	0.0263 *
Hyaluronic acid, ng/mL	27.9 ± 22.5	45.4 ± 45.0		0.982 (0.955–1.008)	0.1708
MDA-LDL, U/L	133.6 ± 47.4	166.1 ± 79.3		0.990 (0.973–1.005)	0.1837
SFA, cm^2^	563.3 ± 158.7	495.2 ± 86.0		1.004 (0.998–1.010)	0.2133
VFA, cm^2^	261.5 ± 76.3	292.2 ± 78.2		0.995 (0.983–1.005)	0.3012
Liver volume, mL	2167.8 ± 506.4	2437.5 ± 712.2		0.999 (0.998–1.001)	0.2349
L/S ratio, no unit	0.7 ± 0.2	0.9 ± 0.1		0.061 (0.0006–2.434)	0.1464
**Multivariate Analysis**					
**Parameters**	**Odds Ratio**		**95% Confidence Interval**		***p*-Value**
HbA1c, %	0.435		0.130–1.040		0.0620
AST, IU/L	0.994		0.964–1.023		0.6756
γ-GTP, IU/L	0.979		0.939–1.001		0.0645
Type-4 collagen 7S, ng/mL	1.411		1.411–6.909		0.6081
L/S ratio, no unit	0.002		<0.001–0.592		0.0310

Values are the mean ± standard deviation. * Parameters with *p* < 0.05. Multivariate analyses were performed using parameters with *p* < 0.15. Abbreviations: NASH, nonalcoholic steatohepatitis; BMI, body mass index; FBG, fasting blood glucose; IRI, immunoreactive insulin; HbA1c, hemoglobin A1c; HOMA-IR, homeostasis model for assessing insulin resistance; AST, aspartate aminotransferase; ALT, alanine aminotransferase; γ-GTP, γ-glutamyl transpeptidase; MDA-LDL, malondialdehyde-modified low-density lipoprotein cholesterol; SFA, subcutaneous fat area; VFA, visceral fat area; L/S ratio, liver/spleen ratio.

**Table 4 biomedicines-10-00453-t004:** Clinical features of patients with non-improved NASH after LSG.

Parameters	Improvement (*n* = 23)	Non-Improvement (*n* = 9)		Odds Ratio (95% Confidence Interval)	*p*-Value
**Postoperative Parameters (12 Months after LSG)**					
Body weight, kg	86.9 ± 16.6	92.7 ± 13.9		0.977 (0.927–1.026)	0.3293
%TWL, %	27.5 ± 7.6	21.8 ± 5.9		1.150 (0.991–1.336)	0.0331 *
FBG, mg/dL	90.5 ± 18.4	101.0 ± 14.9		0.969 (0.918–1.011)	0.1457
IRI, μU/mL	8.5 ± 4.3	15.4 ± 9.9		0.847 (0.683–0.968)	0.0122 *
HbA1c, %	5.5 ± 0.4	6.1 ± 0.7		0.136 (0.017–0.636)	0.0094 *
HOMA-IR, no unit	1.8 ± 1.2	4.2 ± 3.3		0.547 (0.221–0.910)	0.0153 *
AST, IU/L	16.5 ± 4.4	27.7 ± 19.9		0.887 (0.751–0.982)	0.0149 *
ALT, IU/L	15.4 ± 6.8	29.2 ± 11.3		0.840 (0.714–0.936)	0.0004 *
γ-GTP, IU/L	23.8 ± 32.4	86.3 ± 153.5		0.987 (0.956–1.001)	0.0666
LDL-C, mg/dL	101.8 ± 34.1	112.9 ± 33.0		0.990 (0.962–1.015)	0.4137
TG, mg/dL	85.2 ± 43.3	135.9 ± 103.4		0.988 (0.969–1.001)	0.0642
Hyaluronic acid, ng/mL	30.4 ± 18.0	38.4 ± 42.6		0.989 (0.960–1.019)	0.4535
MDA-LDL, U/L	96.9 ± 24.4	123.2 ± 17.9		0.939 (0.882–0.982)	0.0032 *
PAI-1, ng/mL	20.3 ± 13.9	25.1 ± 19.9		0.982 (0.932–1.032)	0.4468
VFA, cm^2^	140.6 ± 79.9	195.7 ± 28.0		0.989 (0.975–1.001)	0.0810
Waist, cm	102.3 ± 13.8	108.8 ± 6.9		0.958 (0.886–1.027)	0.2320
Liver volume, mL	1551.9 ± 291.7	1921.7 ± 300.9		0.996 (0.992–0.999)	0.0071 *
L/S ratio, no unit	1.3 ± 0.2	1.2 ± 0.5		6.745 (0.275–544.575)	0.2461
**Multivariate Analysis**					
**Parameters**	**Odds Ratio**		**95% Confidence Interval**		***p*-Value**
HbA1c, %	0.023		<0.001–9.391		0.2550
ALT, IU/L	0.717		0.178–0.953		0.0081
MDA-LDL, U/L	0.953		0.821–1.055		0.3843
Liver volume, mL	0.990		0.951–1.000		0.0566

Values are the mean ± standard deviation. * Parameters with *p* < 0.05. Multivariate analyses were performed using parameters with *p* < 0.01. Abbreviations: NASH, nonalcoholic steatohepatitis; FBG, fasting blood glucose; IRI, immunoreactive insulin; HbA1c, hemoglobin A1c; HOMA-IR, homeostasis model for assessing insulin resistance; AST, aspartate aminotransferase; ALT, alanine aminotransferase; γ-GTP, γ-glutamyl transpeptidase; TG, triglyceride; LDL-C, low-density lipoprotein cholesterol; MDA-LDL, malondialdehyde-modified low-density lipoprotein cholesterol; PAI-1, plasminogen activator inhibitor 1; VFA, visceral fat area; L/S ratio, liver/spleen ratio.

## Data Availability

The data presented in this study are available on request from the corresponding author. The data are not publicly available due to the duty of ensuring the confidentiality of all patients set by Japanese law; therefore, all data have been anonymized and strictly protected from external network connection.

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
