# Peer review of "The Concept of Indeterminable NASH Inducted by Preoperative Diet and Metabolic Surgery: Analyses of Histopathological and Clinical Features"

_biomedicines, 2022, doi:10.3390/biomedicines10020453_

Round 1

Reviewer 1 Report

The authors evaluated retrospectively 63 patients who underwent laparoscopic sleeve gastrectomy (LSG) and had also a liver biopsy during the surgery. The patients had been on the weight loosing diet preoperatively. However, the patients had not undergone the initial liver biopsy before the onset of the preoperative diet. The length of the preoperative diet phase was not defined, the patients underwent the LSG when the weight loss had achieved more than 5% of the initial body weight. The authors defined the LSG procedure as baseline (Figure 2), but the preoperative diet period had preceded the baseline time point. The authors stratified the patients according to the finding in the baseline (perioperative) liver biopsy in subgroups of NASH, NAFL and indeterminate-NASH patients. Patients with indeterminate-NASH had normal liver enzymes at baseline and they had hepatocyte ballooning and/or liver fibrosis without steatosis in the baseline liver biopsy. These patients cannot be classified as a specific form of NASH with regard to the absence of the real baseline liver biopsy before the preoperative diet period. The formulation of indeterminate-NASH concept is unfortunately false without the knowledge of the clinical and laboratory data before the preoperative diet. The patients with “indeterminate-NASH” were in fact patients with partially resolved NASH after the initial weight loss during the preoperative diet period. Liver steatosis regresses more quickly than fibrosis after a weight loss in a patient with NASH. The regression of steatosis was observed in more than one half of the patients who lost 5–7% of the initial body weight (Romero-Gomez M, et al. J Hepatology 2017; 67: 829–46, Vilar-Gomez E, et al. Gastroenterology 2015; 149: 367–78). Liver enzymes decrease often starts quickly after the onset of low energy diet. These fact support the idea that the patients classified as patients with “indeterminate-NASH” were patients with a partial resolution of NASH after the initial weight loss.

The authors picked a number of valuable data documenting the beneficial effect of LSG on the course of NASH and metabolic syndrome. The paired liver biopsies after 6 and 12 months after the LSG procedure should be considered as the most interesting data. Unfortunately the manuscript has not the standard structure, some results were mentioned already in the section Introduction. The manuscript is very difficult to follow. The abstract does not contain crucial data on the number of liver biopsies performed at 6 and 12 months after LSG, these data are missing also in the figure 2 (flowchart). The tables are complicated and contain also less important data regarding documentation of the beneficial effect of the LSG procedure. The less important data should be presented as an additional document. The manuscript should be rewritten completely, the new manuscript should contain only the data on the course of clinical and laboratory data after the LSG. The concept of “indeterminate-NASH” should be removed from the new manuscript and the patients classified as the patients with “indeterminate-NASH” should be evaluated together with the NASH patients.   

Author Response

January 8, 2022

Biomedicines

Dear. Editors:

We are grateful that you are willing to reconsider our manuscript, and we sincerely appreciate the valuable and constructive comments of the reviewers. We believe that these comments have significantly improved the quality of our manuscript. To address the reviewers’ concerns, we have substantially revised the original manuscript. Our point-by-point responses to the comments are provided below:

The authors evaluated retrospectively 63 patients who underwent laparoscopic sleeve gastrectomy (LSG) and had also a liver biopsy during the surgery. The patients had been on the weight losing diet preoperatively. However, the patients had not undergone the initial liver biopsy before the onset of the preoperative diet. The length of the preoperative diet phase was not defined, the patients underwent the LSG when the weight loss had achieved more than 5% of the initial body weight. The authors defined the LSG procedure as baseline (Figure 2), but the preoperative diet period had preceded the baseline time point. The authors stratified the patients according to the finding in the baseline (perioperative) liver biopsy in subgroups of NASH, NAFL and indeterminate-NASH patients. Patients with indeterminate-NASH had normal liver enzymes at baseline and they had hepatocyte ballooning and/or liver fibrosis without steatosis in the baseline liver biopsy. These patients cannot be classified as a specific form of NASH with regard to the absence of the real baseline liver biopsy before the preoperative diet period. The formulation of indeterminate-NASH concept is unfortunately false without the knowledge of the clinical and laboratory data before the preoperative diet. The patients with “indeterminate-NASH” were in fact patients with partially resolved NASH after the initial weight loss during the preoperative diet period. Liver steatosis regresses more quickly than fibrosis after a weight loss in a patient with NASH. The regression of steatosis was observed in more than one half of the patients who lost 5–7% of the initial body weight (Romero-Gomez M, et al. J Hepatology 2017; 67: 829–46, Vilar-Gomez E, et al. Gastroenterology 2015; 149: 367–78). Liver enzymes decrease often starts quickly after the onset of low energy diet. These fact support the idea that the patients classified as patients with “indeterminate-NASH” were patients with a partial resolution of NASH after the initial weight loss.

The authors picked a number of valuable data documenting the beneficial effect of LSG on the course of NASH and metabolic syndrome. The paired liver biopsies after 6 and 12 months after the LSG procedure should be considered as the most interesting data. Unfortunately the manuscript has not the standard structure, some results were mentioned already in the section Introduction. The manuscript is very difficult to follow. The abstract does not contain crucial data on the number of liver biopsies performed at 6 and 12 months after LSG, these data are missing also in the figure 2 (flowchart). The tables are complicated and contain also less important data regarding documentation of the beneficial effect of the LSG procedure. The less important data should be presented as an additional document. The manuscript should be rewritten completely, the new manuscript should contain only the data on the course of clinical and laboratory data after the LSG. The concept of “indeterminate-NASH” should be removed from the new manuscript and the patients classified as the patients with “indeterminate-NASH” should be evaluated together with the NASH patients.

Response: We thank you and sincerely appreciate your thorough and informative comment on our manuscript. Accordingly, we have addressed your concerns to the best of our abilities and implemented your valuable suggestions.

1.However, the patients had not undergone the initial liver biopsy before the onset of the preoperative diet. The length of the preoperative diet phase was not defined, the patients underwent the LSG when the weight loss had achieved more than 5% of the initial body weight. The authors defined the LSG procedure as baseline (Figure 2), but the preoperative diet period had preceded the baseline time point.

Response: Thank you for your valuable comment. We have added the following sentences in the Results section:

Page 9, lines 228-236: In addition, preoperative diet periods and weight-loss effects were compared. The preoperative diet period of NASH, NAFL, and Ind-NASH groups were 80.1±82.5 days, 158.4±251.0 days, and 61.8±41.2 days, respectively, with no significant differences. The preoperative weight-loss (21.6±12.9 kg vs. 9.6±5.7 kg, P=0.0167) and %TWL (15.3±6.2 % vs. 8.3±4.4 %, P=0.0058) in the Ind-NASH group were significantly better than those of the NASH group. On the other hand, preoperative reduction of liver volume of NASH, NAFL, and Ind-NASH groups was 384.2±388.6 mL, 493.8±557.9 mL, and 383.8±389.2 mL, respectively, with no significant differences. Based on these findings, liver volume equally decreased in each group; however, the residue of liver steatosis was remained uneven.

2.The authors stratified the patients according to the finding in the baseline (perioperative) liver biopsy in subgroups of NASH, NAFL and indeterminate-NASH patients. Patients with indeterminate-NASH had normal liver enzymes at baseline and they had hepatocyte ballooning and/or liver fibrosis without steatosis in the baseline liver biopsy. These patients cannot be classified as a specific form of NASH with regard to the absence of the real baseline liver biopsy before the preoperative diet period.

Response: Thank you for pointing it out. Accordingly, we have added the following sentences in the Discussion section:

Page 22, lines 434-442: We stratified the patients according to the histopathological findings of the intraoperative liver biopsy conducted in NASH, NAFL, and Ind-NASH groups. The true meaning of Ind-NASH should be clarified using paired liver biopsy both during the initial visit and during LSG. However, it should be noted that the cohort of the present study was limited to patients with severe obesity. In severely obese patients, performing a safe ultrasound-guided liver biopsy can be challenging due to thick subcutaneous fat. In addition, major complications such as bleeding and biloma formation should be avoided for conducting the present study and ensure it adheres to the ethical regulations. For these reasons, we chose intraoperative liver biopsy as the initial histopathological evaluation.

3.Liver steatosis regresses more quickly than fibrosis after a weight loss in a patient with NASH. The regression of steatosis was observed in more than one half of the patients who lost 5–7% of the initial body weight (Romero-Gomez M, et al. J Hepatology 2017; 67: 829–46, Vilar-Gomez E, et al. Gastroenterology 2015; 149: 367–78). Liver enzymes decrease often starts quickly after the onset of low energy diet. These fact support the idea that the patients classified as patients with “indeterminate-NASH” were patients with a partial resolution of NASH after the initial weight loss.

Response: Thank you for your valuable and insightful comment. We have added the following sentences in the Discussion section:

Page 21, lines 416-423: Our protocol liver biopsy revealed incremental improvement from NASH to normal liver via Ind-NASH in several patients. Based on these results, we determined that Ind-NASH was intrinsically NASH at the initial visit; however, steatosis was lost due to the weight-loss and metabolic effects of LSG. In addition, some previous studies have demonstrated that liver steatosis improved more rapidly than live fibrosis after weight loss in patients with NASH [21, 22]. However, no or few steatosis (< 5%) was observed in Ind-NASH. Therefore, Ind-NASH requires vanishment of liver steatosis, not only improvement (≥ 5%).

4.The paired liver biopsies after 6 and 12 months after the LSG procedure should be considered as the most interesting data. Unfortunately the manuscript has not the standard structure, some results were mentioned already in the section Introduction. The manuscript is very difficult to follow. The abstract does not contain crucial data on the number of liver biopsies performed at 6 and 12 months after LSG, these data are missing also in the figure 2 (flowchart).

Response: Thank you for pointing it out, and we apologize for not including the important data. Accordingly, we have added the following sentences in the Results section:

Page 14, lines 262-268: Protocol liver biopsy at 6 months after LSG revealed that 4 patients could achieve histopathological NASH remission. On the other hand, patients with Ind-NASH became the second most dominant histopathological finding in contrast to intraoperative liver biopsy. At 12 months after LSG, 11 patients achieved histopathological remission. Furthermore, a normal liver became the first dominant histopathological finding. Histopathological NASH returned to normal livers in 15 patients (42.8%) within 12 months following LSG (Figure 5).

Our liver biopsy protocol is described in the Materials and Methods section as follows:

Page 3, lines 96-99: In patients with histopathological NASH by intraoperative liver biopsy, protocol liver biopsies were performed at 6 and 12 months after LSG. When histopathological remission of NASH was achieved, further protocol liver biopsy was not performed for the same patient.

The numbers of protocol liver biopsies are shown in Figure 5.

5.The tables are complicated and contain also less important data regarding documentation of the beneficial effect of the LSG procedure. The less important data should be presented as an additional document.

Response: Thank you for your valuable feedback. We have deleted some parameters regarding renal function in Table 1. In Tables 3 and 4, parameters with P values > 0.5 have been deleted.

6.The concept of “indeterminate-NASH” should be removed from the new manuscript and the patients classified as the patients with “indeterminate-NASH” should be evaluated together with the NASH patients.

Response: Thank you for your valuable suggestion. We have already compared the fibrosis group (NASH, Ind-NASH) with the non-fibrosis group (NAFL, normal liver) using other populations.

Nikai H, Sasaki A, Umemura A, Takahashi N, Nitta H, Akasaka R, Kakisaka K, Kuroda H, Ishida K, Takikawa Y. Predictive scoring system for advanced liver fibrosis in Japanese patients with severe obesity. Surg Today. 2021 Sep;51(9):1513-1520.

In addition, we have communicated this information to Reviewers 1 and 2, and who agreed with our concept; however, Reviewer 3 suggested that “indeterminable” is more suitable for the intermediate condition between NASH and normal liver. Therefore, we renamed it “indeterminable NASH”.

Sincerely,

Akira Sasaki, M.D., Ph.D.

Department of Surgery, Iwate Medical University School of Medicine,

2-1-1 Idaidori, Yahaba-cho, Shiwa-gun, Iwate, 028-3695 Japan

Reviewer 2 Report

Good job with an in-depth analysis of all aspects of nash (biochemical, imaging, biopsy)
A prospective study could be useful in which the "behavioral aspects" are also analyzed (such as diet, physical exercise, work stress, economic conditions, etc.)

Minor corrections:

  1. bibliography is not cited in the text (at least in my manuscript!!!)
  2. figure 5: 35 biopsies at 6 months must be inserted under 35 nash and not overall
  3. the other 28 biopsies are not shown in the same scheme (all normal?)

Author Response

January 8, 2022

Biomedicines

Dear. Editors:

We are grateful that you are willing to reconsider our manuscript, and we sincerely appreciate the valuable and constructive comments of the reviewers. We believe that these comments have significantly improved the quality of our manuscript. To address the reviewers’ concerns, we have substantially revised the original manuscript. Our point-by-point responses to the comments are provided below:

Good job with an in-depth analysis of all aspects of Nash (biochemical, imaging, biopsy). A prospective study could be useful in which the "behavioral aspects" are also analyzed (such as diet, physical exercise, work stress, economic conditions, etc.)

Response: Thank you for your kind comment and encouraging feedback. We’ll try to investigate the relationships between conditions of NASH and behavioral aspects in near future.

  1. Bibliography is not cited in the text (at least in my manuscript!!!)

Response: Thank you for pointing it out. Accordingly, we have added appropriate citations along with the order of our references.

  1. Figure 5: 35 biopsies at 6 months must be inserted under 35 nash and not overall

Response: Thank you for pointing it out. As instructed, we have revised legends of  Figure 5.

Figure legend 3 has been revised as follows:

Page 14, lines 270-272: Figure 5. Patients flowchart are shown. At 6 months after LSG, only 35 patients with NASH received liver biopsy. Four patients could achieve histopathological remission at 6 months after LSG. Therefore, 31 patients received liver biopsy at 12 months after LSG.

  1. The other 28 biopsies are not shown in the same scheme (all normal?)

Response: Thank you for pointing it out, and we apologize for not including the important data. Accordingly, we have added the following sentences in the Results section:

Page 14, lines 262-268: Protocol liver biopsy at 6 months after LSG revealed that 4 patients could achieve histopathological NASH remission. On the other hand, patients with Ind-NASH became the second most dominant histopathological finding in contrast to intraoperative liver biopsy. At 12 months after LSG, 11 patients achieved histopathological remission. Furthermore, a normal liver became the first dominant histopathological finding. Histopathological NASH returned to normal livers in 15 patients (42.8%) within 12 months following LSG (Figure 5).

Our liver biopsy protocol is described in the Materials and Methods section as follows:

Page 3, lines 96-99: In patients with histopathological NASH by intraoperative liver biopsy, protocol liver biopsies were performed at 6 and 12 months after LSG. When histopathological remission of NASH was achieved, further protocol liver biopsy was not performed for the same patient.

Sincerely,

Akira Sasaki, M.D., Ph.D.

Department of Surgery, Iwate Medical University School of Medicine,

2-1-1 Idaidori, Yahaba-cho, Shiwa-gun, Iwate, 028-3695 Japan

Reviewer 3 Report

 I appreciate the opportunity to review this manuscript. The authors are dealing with the important matter of the “The brand-new concept of indeterminate-NASH inducted by metabolic surgery: Analyses of histopathological and clinical features”. Based on the findings, the clinical application of this study is not enough currently because of small sample size. However, it is an interesting topic for readers, especially the authors describe a novel term about indeterminate-NASH. I recommend revising below-mentioned point is addressed.

Q1: It’s difficult to distinguish indeterminate-NASH from previous chronic hepatitis (resolved hepatitis B infection or alcoholism). Can the authors describe more about the baseline characteristics about virus markers (HBsAg, anti-HCV Ab and anti-HBs IgG), history of alcohol drinking, and daily physical activity?

Q2: The Ind-NASH group is male predominant, lower FBG and TG but more obesity. Dose it can be explained by the effect of metabolic health obesity? Can you explain more about why the Ind-NASH group has higher body weight but less steatosis?

Author Response

January 8, 2022

Biomedicines

Dear. Editors:

We are grateful that you are willing to reconsider our manuscript, and we sincerely appreciate the valuable and constructive comments of the reviewers. We believe that these comments have significantly improved the quality of our manuscript. To address the reviewers’ concerns, we have substantially revised the original manuscript. Our point-by-point responses to the comments are provided below:

I appreciate the opportunity to review this manuscript. The authors are dealing with the important matter of the “The brand-new concept of indeterminate-NASH inducted by metabolic surgery: Analyses of histopathological and clinical features”. Based on the findings, the clinical application of this study is not enough currently because of small sample size. However, it is an interesting topic for readers, especially the authors describe a novel term about indeterminate-NASH. I recommend revising below-mentioned point is addressed.

Response: Thank you for your kind comment and encouraging feedback.

Q1: It’s difficult to distinguish indeterminate-NASH from previous chronic hepatitis (resolved hepatitis B infection or alcoholism). Can the authors describe more about the baseline characteristics about virus markers (HBsAg, anti-HCV Ab and anti-HBs IgG), history of alcohol drinking, and daily physical activity?

Response: Thank you for your valuable questions. Accordingly, we have added the following sentences in the Materials and Methods section:

Page 5, lines 146-149: To exclude other liver diseases, such as viral hepatitis, autoimmune hepatitis, and alcoholic hepatitis, screening laboratory examinations (HBs-antigen, HBs-antibody, HBc-antibody, HCV-antibody, antinuclear antibody, smooth muscle antibody, and immunoglobulin G4) and detailed medical consultation were performed at the first visit.

In addition, we have added the following sentence in the Results section:

Page 6, lines 197-198: There was no patient with other chronic liver diseases, including alcohol abuse, during the initial screening examination.

Q2: The Ind-NASH group is male predominant, lower FBG and TG but more obesity. Dose it can be explained by the effect of metabolic health obesity? Can you explain more about why the Ind-NASH group has higher body weight but less steatosis?

Response: Thank you for your questions. We have added the following sentences in the Discussion section:

Page 21, lines 403-410: Generally, people with sufficient pancreatic β-cell function can accept high calorie input and store them as visceral or ectopic fat deposition [5, 20]; therefore, these patients tend to be heavier than those with normal pancreatic β-cell function. On the other hand, metabolic parameters such as FBG, and TG were lower than patients with NASH because strong β-cell function controls these parameters until insulin secretion is exhausted. Preservation of pancreatic β-cell function results in effective preoperative and postoperative good weight-loss, as previously clarified.

Sincerely,

Akira Sasaki, M.D., Ph.D.

Department of Surgery, Iwate Medical University School of Medicine,

2-1-1 Idaidori, Yahaba-cho, Shiwa-gun, Iwate, 028-3695 Japan

Reviewer 4 Report

Sasaki et al. proposed new concept “indeterminable-NASH” characterized by hepatic inflammation, hepatocyte ballooning, and fibrosis without steatosis. The concept is interesting. Although we often encounter patients with chronic liver disease whose hepatic steatosis <5% but with inflammation and ballooning, it is difficult to make diagnosis. The concept will help us the understanding of NAFLD. I have comments to strength the data.

  1. There were no references in the text. Please add if the authors have a chance to resubmit
  2. P2L65. I thought the authors intended to propose the new concept “indeterminable-NASH” here. However, the authors incorrectly wrote the new concept.
  3. In Figure 1A, the authors showed histological features of Ind-NASH. However, I cannot see inflammation and ballooning in the present magnification. Thus, the authors should add the magnified photo for readers to see inflammation and ballooning.
  4. The authors mentioned liver fibrosis in the legend of Figure 1. Is Ind-NASH characterized by central fibrosis? If the authors have additional data on the distribution of liver fibrosis noted in Ind-NASH, this reviewer strongly recommend that the authors show the data of fibrosis distribution.
  5. The authors used a parameter “liver to spleen (L/S) ratio”. Is the ratio volume? or the density of CT? If the ratio is volume, readers may be confused because the volumes of the spleen were so large.
  6. In Figure 3, the authors mentioned PFS. However, we cannot see the pericellular fibrosis at this magnification. Please add magnified photo.
  7. In Figure 6, the authors may use different magnified photo (The left upper). If so, please add scale bar.
  8. Ind-NASH is an interesting concept (Fig 11). Does this phenomenon specifically observe in patients who received metabolic surgery? Please add discussion. We usually see patients who show improvement in transaminase (inflammation) first, then fatty liver during weight loss by diet and exercise.
  9. The limitation in the present study we have to pay attention is an assessment of fatty change. Although liver biopsy is still a gold standard in diagnosis of NASH, it has sampling error. Although other modalities such as FibroScan and MRI elastography strength the data of present study, these modalities are not available at any institution. However, the authors should mention the limitation of liver biopsy.
  10. P23L460. I think “intermediate” is not correct.

Author Response

January 8, 2022

Biomedicines

Dear. Editors:

We are grateful that you are willing to reconsider our manuscript, and we sincerely appreciate the valuable and constructive comments of the reviewers. We believe that these comments have significantly improved the quality of our manuscript. To address the reviewers’ concerns, we have substantially revised the original manuscript. Our point-by-point responses to the comments are provided below:

Sasaki et al. proposed new concept “indeterminable-NASH” characterized by hepatic inflammation, hepatocyte ballooning, and fibrosis without steatosis. The concept is interesting. Although we often encounter patients with chronic liver disease whose hepatic steatosis <5% but with inflammation and ballooning, it is difficult to make diagnosis. The concept will help us the understanding of NAFLD. I have comments to strength the data.

Response: Thank you for your valuable insights.

1.There were no references in the text. Please add if the authors have a chance to resubmit.

Response: Thank you for pointing it out. Accordingly, we have added appropriate citations along with the order of our references.

2.I thought the authors intended to propose the new concept “indeterminable-NASH” here. However, the authors incorrectly wrote the new concept.

Response: Thank you for your comment. As you mentioned, “indeterminable” is more comprehensible than “indeterminate”. Therefore, we have revised “indeterminate” to “indeterminable”.

3.In Figure 1A, the authors showed histological features of Ind-NASH. However, I cannot see inflammation and ballooning in the present magnification. Thus, the authors should add the magnified photo for readers to see inflammation and ballooning.

Response: Thank you for pointing it out. Accordingly, we have revised Figure 1 including magnified pictures.

Figure legend 1 has been revised as follows:

Page 3, lines 72-79: Figure 1. Histopathological differences between Ind-NASH and typical NASH. Microscopic findings are shown as hematoxylin-eosin, Masson trichrome, and silver reticulin from the left one. A. A typical Ind-NASH patient with a low-power field view. Small bars represent 100 μm. Mild steatosis and mild abnormal fibrosis around central veins are seen. B. Focal ballooning in an Ind-NASH patient (left). Small bar represents 60 μm. Central vein fibrosis without steatosis (right). Small bar represents 200 μm. C. A typical NASH patient. Small bars represent 100 μm. Severe steatosis, hepatocyte ballooning, and periportal fibrosis are seen.

4.The authors mentioned liver fibrosis in the legend of Figure 1. Is Ind-NASH characterized by central fibrosis? If the authors have additional data on the distribution of liver fibrosis noted in Ind-NASH, this reviewer strongly recommend that the authors show the data of fibrosis distribution.

Please refer the answer for question 3.

5.The authors used a parameter “liver to spleen (L/S) ratio”. Is the ratio volume? or the density of CT? If the ratio is volume, readers may be confused because the volumes of the spleen were so large.

Response: Thank you for your questions. We have added the following sentences in the Materials and Methods section:

Page 4, lines 139-143: In addition, we also measured liver volume using CT volumetry. To calculate the liver-to-spleen (L/S) ratio, we measured hepatic and splenic CT attenuation values on non-contrast CT using 20 circular region-of-interest cursors in the liver and spleen. We obtained all measurements in the region of uniform parenchymal CT attenuation.

6.In Figure 3, the authors mentioned PFS. However, we cannot see the pericellular fibrosis at this magnification. Please add magnified photo.

Response: Thank you for pointing it out. As instructed, we have revised Figure 3, including magnified pictures of PFS 1–3.

Figure legend 3 has been revised as follows:

Page 6, lines 173-180: Figure. 3. Histopathological definitions of PFS. PFS 0, no liver fibrosis, and the small bars represent 200 μm. PFS 1, pericellular fibrosis confined to the proximity of central veins and is present in approximately < 50 % of these veins; high-power field view (right). Small bars represent 300 μm and 100 μm (from the left). PFS 2, pericellular fibrosis confined to the proximity of central veins and is present in > 50 % of the central veins; high-power field view (right). Small bars represent 200 μm and 100 μm (from the left). PFS 3, pericellular fibrosis around the central veins with periportal fibrosis or bridging fibrosis; high-power field view (right). Small bars represent 200 μm and 100 μm (from the left).

7.In Figure 6, the authors may use different magnified photo (The left upper). If so, please add scale bar.

Response: Thank you for your suggestion. Accordingly, we have magnified Figure 6 and added scale.

Figure legend 6 has been revised as follows:

Page 15, lines 285-289: Figure 6. Protocol live biopsy confirms sequential histopathological improvement in the patient with good weight-loss effects before and after LSG. BMI at LSG, 6 months after LSG (6POM), and 1 year after LSG (1POY) were 47.2 kg/m2, 43.6 kg/m2, and 40.0 kg/m2, respectively. The patient could achieve histopathological remission of NASH at 12 months after LSG. Small bars represent 50 μm.

8.Ind-NASH is an interesting concept (Fig 11). Does this phenomenon specifically observe in patients who received metabolic surgery? Please add discussion. We usually see patients who show improvement in transaminase (inflammation) first, then fatty liver during weight loss by diet and exercise.

Response: Thank you for your suggestion. We have added the following sentences  in the Discussion section:

Page 21, lines 425-430: Transition from NASH to Ind-NASH may occur from the preoperative phase, and we usually observe that transaminase firstly decreases as the inflammation improves. In addition, a preoperative very low-calorie diet and potential hypersensitivity for weight-loss effects bring dramatic reduction of hepatic fat accumulation [23]. Therefore, we conclude that inflammation and weight-loss occur simultaneously.

9.The limitation in the present study we have to pay attention is an assessment of fatty change. Although liver biopsy is still a gold standard in diagnosis of NASH, it has sampling error. Although other modalities such as FibroScan and MRI elastography strength the data of present study, these modalities are not available at any institution. However, the authors should mention the limitation of liver biopsy.

Response: Thank you for pointing it out. Accordingly, we have added the following sentences in the Discussion section:

Page 23, lines 514-519: Regarding the accuracy of liver biopsy, it has been shown to have a high rate of sampling error because the liver may have both focal and whole histopathological changes due to its volume. To supplement this uncertainty, FibroScan and magnetic resonance imaging may be useful modalities to combine with liver biopsy [37-39]. Nonetheless, we did not apply these modalities in the present study. Therefore, future studies employing a combination of these modalities are warranted.

10.I think “intermediate” is not correct.

Response: We appreciate your valuable suggestion.

Page 23, lines525-526: In this study, Ind-NASH was observed as an intermediate condition between NASH and normal liver; Ind-NASH may represent a recovery process from NASH by metabolic surgery. Therefore, we believe that “intermediate” is accurate.

Sincerely,

Akira Sasaki, M.D., Ph.D.

Department of Surgery, Iwate Medical University School of Medicine,

2-1-1 Idaidori, Yahaba-cho, Shiwa-gun, Iwate, 028-3695 Japan

Round 2

Reviewer 1 Report

I have still the same major comment on the revised manuscript. The absence of liver histology before the preoperative diet period impede formulation of the brand-new concept of indeterminable-NASH inducted by metabolic surgery. I believe that the patients with “indeterminable-NASH” would had had the finding compatible with the diagnosis of a “typical” NASH in the liver biopsy before preoperative diet period, if it had been performed. They showed a significant weight loss on preoperative diet and therefore, their liver histology changed quickly. The fact that they had initially normal ALT activity and NAFIC score is not surprising. Verma S et al. showed (Liver Int., 2013) that among patients with NAFLD and normal ALT a proportion (37.5%) of patients who had liver histology compatible with the diagnosis of NASH. Also NAFIC score had no absolute diagnostic accuracy in the prediction of NASH. The AUROC for predicting NASH was 0.851 in the estimation group and 0.782 in the validation group in the study by Sumida Y, et al. (J Gastroenterol., 2011). 

Author Response

Dear. Editors:

We are grateful that you are willing to reconsider our manuscript, and we sincerely appreciate the valuable and constructive comments of the reviewers. We believe that these comments have significantly improved the quality of our manuscript. To address the reviewers’ concerns, we have substantially revised the original manuscript. Our point-by-point responses to the reviewers’ comments are provided below.

I have still the same major comment on the revised manuscript. The absence of liver histology before the preoperative diet period impede formulation of the brand-new concept of indeterminable-NASH inducted by metabolic surgery. I believe that the patients with “indeterminable-NASH” would had had the finding compatible with the diagnosis of a “typical” NASH in the liver biopsy before preoperative diet period, if it had been performed. They showed a significant weight loss on preoperative diet and therefore, their liver histology changed quickly. The fact that they had initially normal ALT activity and NAFIC score is not surprising. Verma S et al. showed (Liver Int., 2013) that among patients with NAFLD and normal ALT a proportion (37.5%) of patients who had liver histology compatible with the diagnosis of NASH. Also NAFIC score had no absolute diagnostic accuracy in the prediction of NASH. The AUROC for predicting NASH was 0.851 in the estimation group and 0.782 in the validation group in the study by Sumida Y, et al. (J Gastroenterol., 2011).

Response: We thank you and sincerely appreciate your thorough, informative comment on our manuscript. Accordingly, we addressed your concerns to the best of our abilities and implemented your valuable suggestions.

1.The absence of liver histology before the preoperative diet period impede formulation of the brand-new concept of indeterminable-NASH inducted by metabolic surgery. I believe that the patients with “indeterminable-NASH” would have the finding compatible with the diagnosis of a “typical” NASH in the liver biopsy before preoperative diet period, if it had been performed. They showed a significant weight loss on preoperative diet and therefore, their liver histology changed quickly.

Response: Thank you for your valuable comment. We also absolutely agree with your suggestion (page 21, lines 395-398 and 416-418). We have changed the title and the contents of the manuscript according to your suggestion. However, the clinical features and responsibility for preoperative diet and LSG in Ind-NASH patients are quite different from those of patients with NASH. Please understand that these findings are a breakthrough in evaluating the therapeutic effects of not only medical treatments but also MS.

Page 1, lines 2-4: The title of the manuscript was revised form “The brand-new concept of indeterminate-NASH inducted by metabolic surgery: Analyses of histopathological and clinical features” to “The concept of indeterminable-NASH inducted by preoperative diet and metabolic surgery: Analyses of histopathological and clinical features”.

We have added the following sentences in the Introduction section.

Page 2, lines 67-70: In the present study, some patients do not have sufficient steatosis but does have clear inflammation and/or fibrosis that resembles that in NASH before preoperative weight loss. We term these histopathological finding “indeterminable NASH” (Ind-NASH).

Page 3, lines 84-86: Furthermore, we define the term “indeterminable NASH” as a concept of the histopathological condition of returning from NASH to normal liver.

Page 20, lines 388-389: This finding highlights to a notable histopathological concept that has never been deeply discussed.

Page 21, lines 434-444: Most practitioners may assume that patients with Ind-NASH have typical histopathological NASH on the initial visit because all patients are severe obesity and have severe obesity-related diseases. However, we again emphasize that patients with Ind-NASH can achieve good weight-losses not only during the preoperative diet but also during post-LSG periods. We encountered 33 Ind-NASH patients from their intraoperative liver biopsies to 12 months after LSGs. In these patients, postoperative %TWL higher than in NASH-sustained patients (26.1% vs. 20.1%; P = 0.0331). In addition, FBG (92.3 mg/dL vs. 100.1 mg/dL; P = 0.0292), IRI (8.6 μU/mL vs. 14.6 μU/mL; P = 0.0058), and HOMA-IR (2.0 vs. 3.9; P = 0.0048) were significantly lower than those of NASH-sustained patients regardless of whether they had T2D. Patients with Ind-NASH successfully achieved improved insulin resistance and pancreatic β-cell function.

2.The fact that they had initially normal ALT activity and NAFIC score is not surprising. Verma S et al. showed (Liver Int., 2013) that among patients with NAFLD and normal ALT a proportion (37.5%) of patients who had liver histology compatible with the diagnosis of NASH. Also NAFIC score had no absolute diagnostic accuracy in the prediction of NASH. The AUROC for predicting NASH was 0.851 in the estimation group and 0.782 in the validation group in the study by Sumida Y, et al. (J Gastroenterol., 2011).

Response: Thank you for pointing this out. Accordingly, we have added the following sentences in the Discussion section.

Page 23, lines 508-512: Currently, various non-invasive tests for the diagnosis of NASH have been reported [30-33], and the AUCs of these scoring systems are quite accurate [30, 31]. However, we have demonstrated that preoperative scoring systems for NASH do not correctly reflect histopathological findings when applied to severely obese patients [5].

Page 23, lines 520-522: Most non-invasive tests for the diagnosis of NAFLD include AST/ALT levels [33]; however, Verma et al. reported that the AUCs for ALT levels correlating with NASH and advanced fibrosis were 0.62 and 0.46, respectively [39].
